# Bicyclo[3.2.0]carbocyclic Molecules and Redox Biotransformations: The Evolution of Closed-Loop Artificial Linear Biocatalytic Cascades and Related Redox-Neutral Systems

**DOI:** 10.3390/molecules28217249

**Published:** 2023-10-24

**Authors:** Andrew Willetts

**Affiliations:** Curnow Consultancies Ltd., Trewithen House, Helston TR13 9PQ, Cornwall, UK; andrewj.willetts@btconnect.com; Tel.: +44-07-8669-68487 or +44-139-285-136

**Keywords:** artificial linear biocatalytic cascades, bicyclo[3.2.0]carbocyclic molecules, Baeyer–Villiger monooxygenases, alcohol dehydrogenases, prostaglandin synthons, coupled-enzyme membrane reactor

## Abstract

The role of cofactor recycling in determining the efficiency of artificial biocatalytic cascades has become paramount in recent years. Closed-loop cofactor recycling, which initially emerged in the 1990s, has made a valuable contribution to the development of this aspect of biotechnology. However, the evolution of redox-neutral closed-loop cofactor recycling has a longer history that has been integrally linked to the enzymology of oxy-functionalised bicyclo[3.2.0]carbocyclic molecule metabolism throughout. This review traces that relevant history from the mid-1960s to current times.

## 1. Introduction

The long-recognised potential of oxy-functionalised bicyclo[3.2.0]heptenones and the equivalent heptenols to offer different opportunities in each ring for the subsequent stereocontrolled assembly of a significant number of valuable products has been reaffirmed within the last few months [1]. However, this class of fused-ring molecules and their involvement in redox and redox-neutral biocatalysis has a much longer history, including the first reported examples of closed-loop artificial linear cascade reactions [2], hence prompting this review. Given the frenetic pace that characterises the dissemination of much of modern scientific research, it can prove beneficial to take such a longer-term view of the idiosyncratic history that defines the development of research progress in any particular scientific area. Hindsight is always 20:20 vision, and if acknowledged and reviewed constructively can serve as an antidote to history repeating itself.

Prior to the early 1960s, there were no reports in the scientific literature in any context of compounds containing an oxy-functionalised bicyclo[3.2.0]carbocyclic ring system, and interest in such compounds was essentially esoteric. This differs sharply from the explosion of involvement and endeavour [3], driven mainly by the pharmaceutical industry, that followed the discovery and subsequent characterisation of natural products such as penicillin G, originally from *Penicillium notatum* (Figure 1a [4]), and carbapenem from *Serratia* and *Erwinia* spp. (Figure 1b [5]), which proved to be heterocyclic fused-ring compounds containing a signature four-atom β-lactam ring structure. Contrastingly, it was only by pure serendipity that the first synthetic oxy-functionalised bicyclo[3.2.0]carbocyclic fused-ring compound was isolated and identified [6]. While attempting to replicate the earlier recorded synthesis of piperitenone by the self-condensation of geranic acid (Figure 1c), a research team working for the major pharmaceutical company Pfizer in the mid-1960s used a sample of the acid containing residual traces of silver oxide. Their use of the contaminated polyunsaturated fatty acid probably accounts for the detection in the resulting neutral fraction of not only the expected piperitenone, but an additional product that was then isolated and reported to be ‘2,6,6-trimethyl[3.2.0]bicyclo-2-heptene-7-one’ (Figure 1d: current nomenclature = 4,7,7-trimethylbicyclo[3.2.0]hept-3-en-6-one). The bicyclic ketone, the value of which Pfizer saw as a derivatised cyclobutanone, was assumed to be a racemic mixture.

Intrigued by Pfizer’s reported chemical synthesis, researchers at the University of Arizona, in collaboration with du Pont de Nemours, another company with extensive pharmaceutical interests, then initiated a search for equivalent molecules that could be extracted from natural sources, principally terpene-rich plant material. By 1967, this initiative resulted in the detection and identification of both enantiocomplementary isomers of Pfizer’s derivatised cyclobutanone in the form of the natural products D-filifolone (Figure 2a, reported [α]^24^ + 307°) from the Australian citrus bush *Zieria smithii*, and L-filifolone (Figure 2b, reported [α]^25^ − 270°) from the Arizona sand sage *Artemisia filifolia* [7]. While a more recent review of the relevant published literature [8] confirmed the occurrence, mainly in plant species, of a relatively small number (58) of other natural products containing an identifiable bicyclo[3.2.0]heptane signature motif, the majority of these were predominantly tricyclic compounds, with the third ring being ‘fused’ at either the 1-2, 2-3, 1-7, or 6-7 C-C bond, or ‘bridged’ between C2 and C7 of the consensus 7-carbon bicyclic ‘skeleton’. Significantly, other than the original decades-old identification of D- and L-filifolone [7], the only additional natural product reported to contain an ‘unfused’ bicyclo[3.2.0]carbocyclic ring structure was raikovenal (Figure 2c), isolated from the marine ciliate *Euplotus raikovi* found in samples taken both in the Mediterranean [9] and off the Altantic coast of Morocco [10]. However, unlike the enantiomeric filifolones, raikovenal does not contain a cyclobutanone ring.

While neither Pfizer or du Pont de Nemours rated oxy-functionalised bicyclo[3.2.0]carbocyclic ring systems worthy of investigating further, a subsequent sequence of key interrelated developments commencing in the late 1960s served to spark much wider general interest. The initial catalyst was the publication in September 1969 of E.J. Corey’s seminal chemosynthetic route to PGE_2_ and PGF_2α_ [11] which drew attention to the value of a related series of versatile disubstituted bicyclic lactone synthons referred to as ‘Corey lactones’ (Figure 3A). This in turn prompted the development of an alternative route to the Corey lactones (Figure 3B) initiatated by the lactonisation of 7,7-dichlorobicyclo[3.2.0]heptan-6-one [12]. Interestingly, in a case of history repeating itself, Zhu et al. [1] deployed an equivalent enzyme-catalysed initiating step in their chemoenzymetic synthesis of PGF2_α_ almost half a century later (vide infra). The wider potential value of 7,7-dichlorobicyclo[3.2.0]heptan-6-one and the related hept-2-en-6-one was soon recognised [13,14], resulting in the development of chemocatalytic routes to various analogue prostaglandins [15,16,17] and the structurally related thromboxanes [18,19,20] and clavulones [21]. While many of these routes served to generate racemic mixtures of these molecules, it was recognised that significant advantages would result from the further successful development of equivalent homochiral synthetic routes. One possible way to achieve this would be by modifying the established methodologies to incorporate at least one enantiospecific enzyme-catalysed step, thereby facilitating chemoenzymatic access to the equivalent optically pure molecules, which have significant beneficial medicinal implications. The recognition of the strategic value of developing such a biocatalytic initiative is easy to understand, given that the inclusion of equivalent roles for enzymes in the chemoenzymatic synthesis of medicinally important compounds has a long and successful history stretching back several decades. The use of pyruvate decarboxylase present in a whole cell culture of *Saccharomyces cerevisiae* to generate L-phenylacetylcarbinol, a key intermediate in the production of D-ephedrine, provided an early precedent for the power of this technology on an industrial scale [22]. Further, the application of enzymes in synthetic organic chemistry had already been pioneered for a number of years [23], principally by Charles Sih, J. Brian Jones, and David Perlman [24]. Throughout those formative years, a consistent theme had been the consistently recognised value of the exquisite chemo-, and/or regio-, and/or stereospecificity of particular enzymes.

The 40+ year history of the interaction between redox biocatalysis and oxy-functionalised bicyclo[3.2.0]carbocyclic molecules has been highly defined in a number of interrelated ways. A principal driver for this was that the relevant formative research was pioneered by a relatively few preeminent scientists, principally Roger Newton, Roland Furstoss, and Stanley Roberts. Further, the research has been restricted to a relatively small number of different enzymes, dictated principally by the oxy-functionalised nature of the targeted systems. Commencing in 1979, Roger Newton pioneered a decade of research that deployed a small number of different alcohol dehydrogenases (ADHs: EC 1.1.1.x) as useful redox enzymes able to bioreduce bicyclic secondary ketones to the corresponding alcohols. However, from the late 1980s onwards, initiatives by Roland Furstoss and Stanley Roberts promoted ketone-utilising Baeyer–Villiger monooxygenases (BVMOs) to become the predominantly used biocatalysts for the biotransformation of oxy-functionalised bicyclic molecules. Significantly, BVMOs, which are NAD(P)H-dependent biooxygenating enzymes, are redox complementary biocatalysts to NAD(P)^+^-dependent ADHs functioning biooxidatively (vide infra). The source of competent ADH and BVMO enzymes has also been highly defined. Contrastingly, while the known natural products containing a relevant oxy-functionalised bicyclic fused-ring structure have all been sourced from plant material [7,8], there are no corresponding records of the concerted screening of plant ADH or BVMO enzymes for activity with abiotic bicyclo[3.2.0]carbocyclic molecules. With the exception of horse liver alcohol dehydrogenase (HLADH), all the other ADH and BVMO enzymes deployed to date have been sourced from either eukaryotic (fungi and yeasts) or prokaryotic (bacteria) microorganisms. This probably reflects the acknowledged cathodic metabolic activities of many microorganisms, a characteristic resulting principally from a combination of some economy of genetic control (two or more inducers able to regulate single gene) plus some economy of genetic material (a range of substrates metabolised by the enzyme coded for by a single gene). This was a proposal first advanced following the pioneering studies by Irwin Gunsalus, Peter Chapman and Nick Ornston [25,26].

When considering the use of any enzyme for biocatalysis, three alternative basic strategies can be deployed. Depending on the nature of the particular targeted activity, either a whole cell preparation (growing culture; resting or immobilised cells), a fractured cell preparation, or one or more isolated enzymes (aqueous or organic suspension; immobilised suspension) have been deployed. The merits and demerits of each of these options have been extensively reviewed [27,28,29,30]. Of particular significance with respect to the involvement of ADHs and BVMOs in bicyclo[3.2.0]carbocyclic fused-ring biotransformations is that both enzyme types are redox biocatalysts, being cofactor-dependent. This introduces both cost and effectiveness considerations when deploying both ADHs and BVMOs as isolated enzymes. These can be mitigated to some extent by deploying relevant cofactor recycling strategies. Cofactor recycling was first introduced for yeast alcohol dehydrogenase (YADH) in the late 1950s [31] by using a coupled enzyme plus an additional sacrificial substrate approach (Figure 4). Subsequently, throughout the 1960s, alternative coupled-substrate methods became predominant [32,33,34] before coupled-enzyme methods returned to predominate from the 1980s onwards. While some of these coupled-enzyme systems, like Levy et al. [31], deployed glucose dehydrogenase [35,36], others depended on glucose-6-phosphate dehydrogenase [37,38], phosphite dehydrogenase [39,40], an EC 1.4.1.x-type amino acid dehydrogenase [41,42], formate dehydrogenase (including genetically engineered NAD^+^ → NADP^+^-dependent variants [43,44,45,46,47,48]), or even hydrogenase [49,50]. These, and other options have been reviewed comprehensively [51]. Examples directly relevant to the specific redox biocatalysis of particular bicyclo[3.2.0]carbocyclic molecules by competent ADHs and BVMOs will be detailed in context. 

## 2. Biotransformations of Oxy-Functionalised Bicyclo[3.2.0]carbocyclic Molecules by ADHs Serving as Single-Step Biocatalysts

ADHs are very widely distributed enzymes in nature [27,28]. A state-of-the-art review comprehensively covering both structural and functional aspects of ADHs and ADH-dependent catalysis has been published very recently [52]. With specific respect to the biocatalysis of various oxy-functionalised bicyclo[3.2.0]carbocyclic ring substrates, commencing in the late 1970s, a number of different ADHs acting in isolation have been tested functioning in either a reductive or an oxidative mode. Three generalisations about these enzymes are relevant. Firstly, the pH range over which they function effectively is quite narrow (pH 7.0–8.0), with the reductive mode being favoured <pH 7.5, and the oxidative mode >pH 7.5 [53,54]: while some ADHs exhibit changes in stereospecificity across their operating range, others such as HLADH do not [55]. Secondly, in general they function more favourably and with higher stereoselectivity when catalysing carbonyl reduction reactions [54,56]. Thirdly, the oxidation of a secondary alcohol results in the destruction of a chiral centre, whereas the corresponding carbonyl reduction reaction can generate either an (*R*)- or (*S*)-alcohol depending on the facial selectivity of hydride delivery [57,58]. Studies conducted in vivo in the 1960s by Vladimir Prelog with cultures of *Curvularia falcata* resulted in ‘Prelog’s rule’ [59], a predictive model for ADH reductions (Figure 5) that corresponds well with the actual outcomes recorded for a number of ADHs subsequently tested in vitro, including YADH and HLADH. However, more recent research has led to the discovery of a number of other so-called anti-Prelog ADHs that catalyse carbonyl reductions with the opposite stereospecificity. Of particular relevance to (*rac*)-bicyclo[3.2.0]ketones, the conformational effect of the bridgehead carbon atoms can be overcome and even reversed by bulky substituents in the adjacent 7-position [60,61], in concurrence with Prelog’s rule.

With the exception of HLADH, interest in deploying ADHs for biocatalysis of oxy-functionalised bicyclo[3.2.0]carbocyclic molecules has focused on microbial enzymes. While there is a very broad spectrum of the characterised microbial alcohol dehydrogenases from both prokaryotic and eukaryotic species, the majority are long-chain (approximately 350 amino acid residues) Zn plus NAD(H)- or NADP(H)-dependent enzymes [62,63,64,65]. Historically, for ease of use, industrial-grade bakers’ or brewers’ yeast replete with one or more NAD(H)-dependent alcohol dehydrogenase isoenzymes (YADH [66,67]) were often deployed. Also frequently tested were various readily available commercial brands of part-purified microbial ADHs, including the NADP(H)-dependent enzyme(s) from *Thermoanaerobium brockii* (TBADH [68,69,70,71,72]), and the NAD(H)-dependent 3α,20β-hydroxysteroid dehydrogenase from *Streptomyces hydrogenans* (HSDH [73,74]). However, surprisingly little use has been made of commercial brands of NAD(H)-dependent HLADH [75,76]. Another influential factor in determining which ADHs have been tested has been their established activity with other oxy-functionalised substrates. The potential applicability of YADH, TBADH, HSDH and HLADH is well recognised in this respect (Table 1). However, some other less widely used microbial alcohol dehydrogenases can also serve potentially valuable roles in this respect. The surprisingly wide confirmed substrate range of both cyclopentanol dehydrogenase (CPDH) from *Pseudomonas* sp. NCIB 9872 (*Comomonas* sp. NCIMB 9872) and cyclohexanol dehydrogenase (CHDH) from *Acinetobacter* sp. NCIB 9871 (*Acinetobacter calcoaceticus* NCIMB 9871) to include a wide range of alicyclic, monocyclic, bicyclic and even polycyclic alcohols (Figure 6 [77]) has proved particularly useful for some relevant biotransformations of bicyclo[3.2.0]hept-2-ene-6-ols (vide infra). With respect to the potential for biotransformation of candidate bicyclo[3.2.0]carbocyclic molecules, there are no directly relevant precedents, although a predictive outcome model has been developed for the bioreduction of substituted cyclohexanones by HLADH (Figure 6 [78]), which was itself developed from the earlier corresponding so-called diamond lattice [79] and cubic space [80] iterative models.

The first ever reported example of an ADH-catalysed biotransformation of an oxy-functionalised bicyclo[3.2.0]carbocyclic molecule, indeed any cyclobutanone, resulted from a study initiated by the then Glaxo Group Research in 1979. They reported that a culture of a commercial brand (Fermipan^®^) of bakers’ yeast (*Saccharomyces cerevisiae*) grown for 24 h in a simple glucose-based medium reduced in excess of 65% of added *(rac*)-bicyclo[3.2.0]hept-2-en-6-one to a 2.2:1 mixture of the corresponding (*6S*)-*endo*- and (*6S*)-*exo*-alcohols (Figure 7 [81]). The configuration of the recovered diastereoisomeric alcohols was established by chemical oxidation into the corresponding known 2-oxa-lactones. Samples of both recovered alcohols were also treated with N-bromosuccinamide to yield the corresponding complementary enantiomeric bromohydrins, analysis of which suggested that the enantiomeric purities of the corresponding *endo*- and *exo*-alcohols were 88% and 84%, respectively. In turn, both bromohydrins could then serve as synthons for the subsequent synthesis of the nature-equivalent prostaglandin PGF2_α_ by two established enantiomer-specific routes [82,83]. Although this overall chemoenzymatic strategy achieved a successful outcome, it was stymied by two principal issues. Firstly, the two diastereoisomeric (*6S*)-alcohol products were not easily separable, and secondly the (*rac*)-ketone proved to be toxic to the bakers’ yeast used when added to the culture >6 g L^−1^.

Newton et al. referred specifically to the novel nature of what they termed the ‘substrate non-specific and product enantioselective bioreduction of bicyclo[3.2.0]hept-2-en-6-one using Bakers’ yeast’. This was because the outcome contrasted directly with the only known precedents, which had recently reported the bioreduction of racemic α- and β-keto esters and cyclohexanones to corresponding homochiral alcohol products by growing cultures of *S. cerevisiae* [84,85]. Although no proposed explanation was advanced at the time, a subsequent programme of research conducted by Glaxo with the fungus *Mortierella ramanniana* resulted in the suggestion that such enantiodivergent outcomes could result from the coincident activity of two (or more) different ADHs in the same microorganism (vide infra). It may also be relevant in this context that a subsequent outcome from the *S. cerevisiae* genome project [86] has confirmed the existence of genes coding for seven different isoenzymes of YADH (ADH1–7), each with its own idiosyncratic sequence identity and corresponding 3-D structure.

Subsequently, Glaxo Group Research intensified efforts to capitalise of the production of the corresponding diastereoisomeric (6S)-alcohols resulting from the bioreduction of (*rac*)-bicyclo[3.2.0[hept-2-en-6-one by Bakers’ yeast by first developing successful chemocatalytic routes to the prostaglandins E_2_ and F_2α_ [87] and then prostaglandin A_2_ [88]. As a consequence of the various favourable bio- and chemocatalytic outcomes achieved to date, Glaxo extended the programme of research to include a number of interrelated strategic objectives [89], firstly to establish whether growing the Fermipan^®^ yeast culture beyond 24 h promoted either any change in the total yield and/or the ratio of the two enantiomeric alcohols produced. Outcomes from this part of the study confirmed that no significant difference in either parameter was found by monitoring the growing Fermipan^®^ yeast culture over 168 h. The second more comprehensive objective was to screen a number of different yeasts and fungi strains held by Glaxo to in order to identify any strain(s) that could selectively bioreduce (*rac*)-bicyclo[3.2.0]hept-2-en-6-one to a single (*6S*)-alcohol of very high enantiomeric purity, which would be comparatively easy to separate from any residual ketone. Included in this screen were six in-house strains of *S. cerevisiae* and a number of other eukaryotic microorganisms. Analysis of the spent medium after growth of each culture for 24 h in the presence of (*rac*)-bicyclic ketone produced two valuable outcomes (Table 2).

Firstly, while the yeast *Rhodotorula ruba* C1768 gave a similar outcome to Fermipan^®^ of bakers’ yeast [81], the yield of both diastereisomeric 6-alcohol products was significantly higher. Secondly, the fungi *Curvularia lunata* C2100 and *Mortierella ramanniana* C2506 both generated relatively high yields of exclusively 6-*endo*-(*1S*,*5R*,*6S*)-bicyclo[3.2.0]hept-2-en-6-ol. However, both the time course of fused-ring ketone reduction and the progressive percentage conversion by the two fungi were different (Table 3). Furthermore, *M. ramanniana* C2506 exhibited better tolerance to increasing levels of the lipophilic fused-ring ketone added into the growth medium.

On the basis of these outcomes, Glaxo chose *M. ramanniana* as the better candidate available at the time for further development of their single enantiomer strategy deploying growing cultures as in vivo biocatalysts. Influential in that decision was an expressed concern [89] that the bioreductive nature of the relevant enzyme(s) would involve in vitro coenzyme recycling, which, although under active development [45], involved material costs considered to be prohibitively high. After further researching the effect of both temperature and aeration on growth-dependent bioreduction of the fused-ring ketone by *M. ramanniana* C2506, a 5-L pilot-scale process for the production of 6-*endo*-(*1S*,*5R*,*6S*)-bicyclo[3.2.0]hept-2-en-6-ol (1 kg wet weight pregrown fungus; 25 g ketone; aeration 1 L min^−1^; 25 °C) was set up, run for 20 h, and the spent growth medium then analysed. Initial h.p.l.c assessment indicated a total of 12.5 g of the 6-*endo*-alcohol had been produced, representing a 50% recovered yield from the relevant ketone starting material. Following subsequent workup and purification, analysis confirmed the isolation of 7.0 g of 6-*endo*-(*1S*,*5R*,*6S*)-bicyclo[3.2.0]hept-2-en-6-ol (e.e. 89%), and 3.4 g of the initial (+)-(*1R*,*5S*)-ketone substrate (e.e. 84%).

Glaxo then sought to capitalise further on the lessons learnt from the *M. ramanniana*-dependent fermentation-based system. The principal aim was to develop a more general method of resolution of additional related (*rac*)-bicyclo[3.2.0]carbocyclic substrates able to deliver chiral synthons of value not only for prostaglandins, but other potential products of interest and value, including boonein [90], huristic acid [91], mulifidene [92], pentalene [93], pentalenolactones [94], ophiobolin [95], and prostacyclin [96]. The chosen direction of travel was influenced principally by the insight of Stanley Roberts, who recognised the potential of developing a multi-stage programme, the initial aims of which were to exploit relevant biocatalytic activities of commercially available isolated alcohol dehydrogenases allied to appropriate coupled-substrate or coupled-enzyme methodologies to promote effective coenzyme recycling in water–organic solvent systems [97]. Subsequent initiatives would then explore the use of immobilised enzyme technology, as already proven effective for ADHs [98,99,100], with the ultimate aim of developing an efficient continuous process that could be operated at scale. The three ADHs initially tested [60], each selected because of their reported broad substrate range, were NAD(H)-dependent HSDH, [73,74]) NAD(H)-dependent HLADH [75,76], and the corresponding NADP(H)-dependent enzyme TBADH [68,69,70,71,72]). The choice of the *S. hydrogenase* HSDH enzyme for testing resulted principally from prior collaboration between Stanley Roberts and Giacomo Carrea at Milan University, as a result of which immobilised preparations of other hydroxysteroid dehydrogenases had been confirmed to retain >60% of initial bioreductive activity after two months in a continuously operated two-phase system [101]. The tested ketone substrates were (*rac*)-bicyclo[3.2.0]hept-2-en-6-one and (*rac*)-7,7-dimethylbicyclo[3.2.0]hept-2-en-6-one. Whereas HLADH and TBADH were tested using coupled-substrate coenzyme recycling with ethanol and propan-2-ol single-phase systems respectively, two alternative aqueous coupled-enzyme systems (glucose:glucose dehydrogenase and ethanol:HLADH) were piloted with HSDH. Each tested biotransformation was allowed to run until 10–20% of the added ketone had been depleted. While no yield data were given for any the tested biotransfomations, the limited outcomes that were reported presented a mixed picture (Table 4). The least promising enzyme proved to be HLADH, which was inactive with the dimethyl-substituted ketone, suggesting some element of comparability with Lemiere et al.’s predictive HLADH model (Figure 6 [78]). Conversely, this enzyme generated exclusively 6-*endo*-alcohol but with low optical purity (<10% e.e.) predominantly from the (-)-(*1S*,*1R*) unsubstituted ketone. TDADH was also inactive with the dimethyl-substituted ketone, but more promisingly did generate exclusively 6-*endo*-alcohol in high optical purity (>95% e.e. [α]_D_ + 69°) almost exclusively from the (-)-(*1S*,*1R*) unsubstituted ketone. While HSDH, like HLADH, performed poorly with the unsubstituted ketone (6-*endo*-alcohol, <10% e.e., predominantly from the (-)-(*1S*,*1R*) unsubstituted ketone), it was the only tested ADH to generate product from the dimethyl-substituted ketone. An equivalent single-product outcome was reported from both coupled-enzyme methods tested, which on analysis proved to be almost optically pure (*6S*)-*endo*-alcohol (>95% e.e.), derived almost exclusively from the corresponding (-)-(*1S*,*1R*) ketone. The highly stereoselective outcomes from the bioreduction of the unsubstituted ketone by TBADH and the dimethyl-substituted ketone by HSDH suggested that both enzymes transfer the pro-(*S*) hydrogen from the relevant reduced nicotinamide cofactor to the *Re*-face of the corresponding carbonyl group. The TBADH outcome was interesting in that prior studies with the same enzyme had indicated that it reduced short-chain aliphatic ketones on the *Si*-face [72]. However, the *Re*-facial selectivity of TBADH with bulky bicyclic ketones was subsequently confirmed independently with both bicyclo[3.2.0]hept-2-en-6-one and bicyclo[3.3.0]oct-2-en-8-one [102].

For both continuity and comparative purposes, growing cultures of *M. ramanniana* were also challenged with the dimethyl-substituted fused-ring ketone, both at 5 g L^−1^ as previously [60], and additionally at 1 g L^−1^. While the rate of reduction of the dimethyl-substituted substrate at both tested levels was lower (<70%) then equivalent to previously recorded rates for the unsubstituted ketone [89], significantly different outcomes were recorded depending on the level of addition of the substituted ketone to the fungus. At the highest tested level (5 g L^−1^), the only detected product was the corresponding 6-*endo*-alcohol, although no relevant yield or chirality data were given. However, the introduction of the dimethyl ketone at 1 g L^−1^ resulted in a different outcome, with a 50% total yield of equal quantities of (-)-(*6S*)-*endo*-alcohol (>95% e.e., [α]_D_ − 145°) and (+)-(*6S*)-*exo*-alcohol (>95% e.e., [α]_D_ + 109°). The absolute configurations of both the alcohols were established by conversion into corresponding lactones of known configuration (Figure 8 [103]), outcomes that concur with Prelog’s rule concerning the selectivity of dehydrogenase enzymes [104]. It was speculated that these contrasting outcomes may reflect the involvement of (at least) two different ADHs, one of which is inhibited by introducing 5 g L^−1^ of the disubstituted ketone to growing cultures of the fungus, a proposal for which they cited equivalent precedents [105,106].

Encouraged by the outcome recorded for the highly enantioselective biotransformation of the 7,7-dimethyl-substituted bicyclic ketone by the HSDH-HLADH single-phase coupled-enzyme system [60], Glaxo expanded the approach to undertake a briefly outlined programme of research [107] aimed at generating equivalent chiral synthons to capitalise on the proven chemocatalytic value of 7-*endo*-chloro- and 7,7-dichlorobicyclo-[3.2.0]hept-2-en-6-ones [108,109,110]. Initially, replicates of the HSDH-HLADH single-phase coupled-enzyme system as developed previously were challenged separately with the two halogenated ketones. While no overall specific yield figures are given, the 7-*endo*-chloroketone yielded exclusively the corresponding 6-*endo*-alcohol (82% e.e., [α]_D_ − 190°).

The recovered unreacted haloketone was dechlorinated and confirmed as (*1S*,*5R*)-bicyclo[3.2.0]hept-2-en-6-one, thereby establishing that the major enantiomer of the chlorohydrin was (*1R*,*5S*)-7-*endo*-chlorobicyclo[3.2.0]-hept-2-en-6-*endo*-ol. An equivalent unquantified outcome was recorded for the dichloroketone, which yielded recovered (*1R*,*5S*)-7,7-dichlorobicyclo[3.2.0]hept-2-en-6-*endo*-ol and unreacted (*1S*,*5R*)-dichloroketone. In a further development, both HSDH and HLADH were co-immobilised using Eupergit-C and then tested in a water–ethanol single-phase system with each of the halogenated test ketones. While in both cases this did result in a small loss of specificity of the recovered halohydrin products (*1R*,*5S*)-7-*endo*-chlorobicyclo[3.2.0]hept-2-en-6-*endo*-ol (80% e.e) and (*1R*,*5S*)-7,7-dichlorobicyclo[3.2.0]hept-2-en-6-*endo*-ol (95% e.e.), the deployment of the solid support simplified the relevant workup procedures and increased the long-term stability of both ADHs. Finally, while no corresponding data were presented, directly comparable coupled-enzyme biotransformation conducted in water-immiscible co-solvent systems (hexane or octane at 30% *v/v* in water) were reported to ‘maintain respectable rates of reduction’.

The progress made with the HSDH-based initiative prompted Glaxo to then expand the research programme to include both additional ketone substrates and multi-phase systems [61], albeit with the exchange of YADH for HLADH as the coupling enzyme for cofactor regeneration. HSDH was challenged both in an initial single-phase coupled-substrate screen with ethanol but no coupling enzyme, and a larger-scale single-phase coupled-enzyme screen with ethanol plus YADH to establish characteristics of the reduction of both established and previously untested ketones. As a preliminary, the effectiveness of HSDH in each of the solvent systems was confirmed by testing the biotransformation of cortisone, a known substrate for this enzyme [111]. Both the relevant affinity (Km) and enantioselectivity (e.e.% outcomes suggested that 7,7-dichloroheptenone was a particularly favoured substrate (Table 5) for HSDH. Also clearly evident from the data was that electron-withdrawing groups increased and electron-donating groups decreased, the reactivity of the keto-function of the bicycloheptenones. Attention was then redirected to examining various multi-phase systems. The most comprehensive data reaffirmed the previously reported preliminary outcome [107] that while HSDH and YADH functioned effectively as a single-phase coupled-enzyme system to bioreduce the favoured 7,7-dichloro-substituted ketone, the introduction of five different organic solvents to create corresponding two-phase systems resulted in each case in a lower rate of bioreduction (Table 6). In another less well-defined two-phase system, an aqueous phase comprising HSDH plus NADH with the 7,7-dichloro-substituted ketone was combined with the cationic detergent cetyltrimethylammonium bromide (CTAB), hexan-1-ol and octane to generate reverse micelles. However, this resulted in a poor outcome (*K_m_* increased nearly 60-fold) compared to the corresponding single-phase coupled-substrate biotransformation. Even less promising, an equivalent two-phase system comprising reverse micelles contructed with the anionic detergent Aerosol OT proved to be totally inactive. Finally, in an attempt to capitalise on the earlier successful deployment of other equivalent immobilised enzymes by Carrea et al. [101], 1.95 g of dichloro-substituted ketone in a water–octanol two-phase system was recycled (3 × 24 h) through a column of co-immobilised HSDH and YADH enabling the subsequent recovery of 185 mg (19% conversion) of the corresponding (*6S*)-*endo*-alcohol.

Attempts to extend the operating window of this system beyond 72 h resulted in a significant decrease in bioreductive activity because of stability issues with YADH. An equivalent unquantified outcome was also reported with the 7-monochloro-substituted ketone.

With hindsight, it is evident that the next relevant publication from Glaxo Group Research [112] served to curtail the company’s involvement with the ADH-based bicyclo[3.2.0]hept-2-en-6-one biotransformation initiative. This finale detailed a manifold approach to the chemoenzymatic synthesis of the lactone (+)-eldanolide, the attractant pheromone of the sugar-cane borer *Eldana saccharina*. Capitalising on the successful outcomes from previous research (Figure 8 [103], Butt et al. initially used three 7-litre growing cultures of *M. ramanniana* to bioreduce a total of 15 g (=0.7 g L^−1^) of (*rac*)-7,7-dimethyl-substituted bicyclic ketone to yield similar amounts of the corresponding (-)-(*6S*)-*endo*-alcohol (3.8 g, recovered yield = 25%; [α]_D_ = −95°; e.e. 84%) and (+)-(*6S*)-*exo*-alcohol (3.4 g, recovered yield = 23%; [α]_D_ = +119°; e.e. >97%). Because of the disappointingly low optical purity of the sought (-)-(*6S*)-*endo*-alcohol, they then instead used the previously proven HSDH-HLADH coupled-enzyme system in aqueous ethanol (Table 4) to successfully reduce 1 g of the (*rac*)-dimethyl-substituted alkenone to yield exclusively the (-)-(*6S*)-*endo*-alcohol (0.22 g, recovered yield = 22%; [α]_D_ = −112°; e.e. >95%). An initial chemocatalytic step was deployed to convert this highly pure (*1R*,*5S*,*6S*)-alcohol to the corresponding (*1R*,*5S*)-ketone (Figure 9), followed by a combination of further chemo- and photocatalytic steps to yield enantiomerically pure (+)-(*3S*,*4R*)-eldanolide ([α]_D_ + 20.4°; corresponding literature value = +20.4° [113]). Significantly, the subsequent curtailment of Glaxo’s involvement with ADH-based fused-ring biocatalysis coincided with the departure from the company of its principal proponent, Stanley Roberts. He left the company to take up a chair at the University of Exeter, thereby enabling him to establish an interdisciplinary research programme to further his joint interests in redox biocatalysis and bicyclo[3.2.0]carbocyclic molecules. This then led directly to pioneering studies that delivered the first reported one-pot artificial biocatalytic linear cascades operating closed-loop cofactor recycling, biocatalytic cascades able to deliver valuable synthons from bicyclo[3.2.0]carbocyclic molecules (vide infra).

Subsequently, in the post-Glaxo era, there have been a limited number of ADH-based publications relevant to redox biocatalysis of oxy-functionalised bicyclic heptenes. These have resulted exclusively from a related series of studies initiated in the mid-1990s by researchers at the University of Ferrara. The initiative was sparked by the recognition that ADHs functioning as single-step biocatalysts could serve either bioreductive or biooxidative roles with (*rac*)-bicyclo[3.2.0]alkenones or alkenols, respectively, to generate versatile synthons for the subsequent chemical elaboration of natural products. As recognised by others [108,109,110], such bicyclic fused ring systems offer different functionalities in each ring suitable for the subsequent stereocontrolled chemocatalytic assembly of a number of natural products, including grandisol, lineatin, and filifolone [114]. Having gained some initial experience of ADHs by using growing cultures of *Bacillus stearothermophilus* ATCC 2027 to kinetically resolve various 1-aryl ethanols [115], the same biocatalyst was then deployed to resolve various racemic bicyclic alcohols, including (*rac*)-6-*endo*-bicyclo[3.2.0[hept-2-en-6-ol [116]. Run on an analytical scale, 10 mg of (*rac*)-*endo*-alcohol in DMF (0.1 mL) was added to 1 litre of growing culture and placed in a reciprocal shaker at 39 °C. After 90 min of incubation, the recovered resolved products were generated: (*1S*,*5R*)-ketone (yield 41%; e.e., 99%) and residual (*1R*,*5S*,*6R*)-*endo*-alcohol (yield 53%; e.e. 80%). Extending the incubation period to 180 min resulted in changes to both the yield and chiral purity of both the recovered (*1S*,*5R*)-ketone (yield 52%; e.e., 70%) and (*1R*,*5S*,*6R*)-*endo*-alcohol (yield 40%; e.e. >99%). A preparative scale run that introduced 0.5 g of the (*rac*)-alcohol in 2.5 mL DMF into 200 mL of growing culture followed by 180 min of incubation at 39 °C was claimed to result in an equivalent outcome, although no supporting data were presented. Encouraged by these initial outcomes, and aware of the reported increase in the enantioselective oxidation of 1-arylethanols by immobilised preparations of *Geotrichum candidum* suspended in hexane [117], a sample of recovered cells from a 48 h culture of *B. stearothermophilus* was first immobilised in an unspecified matrix, then resuspended in 10 ml of heptane containing 40 mg of (*rac*)-6-*endo*-bicyclo[3.2.0]hept-2-en-6-ol and stirred vigorously at 39 °C [118]. Equivalent samples of unimmobilised cells resuspended in either water or heptane were similarly exposed to 40 mg of the (*rac*)-*endo*-alcohol. Subsequent analysis (Table 7) indicated that the (*rac*)-*endo*-alcohol was biooxidised and kinetically resolved when directly suspended in heptane to optically pure (*1S*,*5R*)-bicyclo[3.2.0]hept-2-en-6-one in 49% yield. While the yields of recovered ketone from both the immobilised cells resuspended in heptane and the cells directly resuspended in water were lower, the optical purity was equally impressive in both cases. With all three tested samples, recovered cells were confirmed to be still viable, whereas other tested solvents, including toluene, diethylene glycol, dimethyl ether, and t-butyl methyl ether, were unsuccessful, as they resulted in cell death within 30 min.

In a collaborative partnership with researchers at the University of Bologna, attention then expanded to investigating the bioreduction and biooxidation, respectively, of an expanded range of racemic methyl-substituted bicyclo[3.2.0]hept-en-3-en-6-ones and equivalent -6-ols. The experience of the Bologna scientists with various enantiomerically pure bicyclic alcohols [119,120,121,122,123] was considered valuable in this respect. Capitalising on the prior outcomes of the Glaxo research programme [81,89], the bioreduction reactions were undertaken with growing cultures of bakers’ yeast (source not specified), whereas based on prior experience [115], the biooxidations were conducted with growing cultures of the *B. stearothermophilus* [124,125]. The bicyclic ketone tested with the yeast were (*rac*)-4-methyl- and (*rac*)-1,4-dimethylbicyclo[3.2.0]hept-2-en-6-one, whereas the corresponding (*rac*)-alcohols served as the tested substrates for biooxidation by *B. stearothermophilus*. Preparative-scale bioreductions of the tested ketones (0.5 g ketone, 100 g yeast in 500 mL glucose solution) resulted in both cases with the recovery of corresponding pairs of diastereoisomeric (*6S*)-*endo*- and (*6S*)-*exo*-alcohols in moderate yields, but notably with very high enantiomeric excesses (Table 8).The complementary preparative-scale biooxidations were conducted with 0.5 g aliquots of each (*rac*)-6-*endo*- and (*rac*)-6-*exo*-alcohols introduced into 48 h growing cultures of *B. stearothermophilus* which were then incubated for a further 48 h at 39 °C. Analysis of the spent growth media from both (*rac*)-6-*endo*-alcohol biotransformations confirmed that in each case, the isolated products were the corresponding (*1R*,*5S*)-ketone in good yield and moderate enantiomeric excess and and the corresponding recovered (*1S*,*5R*,*6R*)-heptanol in high enantiomeric excess (Table 8). These recovered unreacted (*6R*)-alcohols had the opposite configuration to the equivalent (*6S*)-alcohols generated by the bakers’ yeast bioreductions conducted by Glaxo Group Research several years previously [81]. Contrastingly, both introduced (*rac*)-6-*exo*-alcohols remained unreacted 48 h after initial inclusion in the growth medium of *B. stearothermophilus*, and continued thus throughout a further 96 h at 39 °C. A directly equivalent negative outcome was briefly reported by Glaxo Group Research over a decade previously when a growing culture of *M. ramanniana* was challenged with (*rac*)-6-*exo*-bicyclo[3.2.0]hept-2-en-6-ol [89].

Subsequently, a crude cell-free extract, termed BSADH, was prepared from a 48 h growing culture of *B. stearothermophilus* [126] and tested without further treatment for the ability to biooxidise the unsubstituted (*rac*)-6-*endo*-alcohol supplemented with NAD^+^ and lactate dehydrogenase plus pyruvate as a suitable coupled-enzyme cofactor recycling system. Additionally, the BSADH extract was tested for the ability to bioreduce the corresponding unsubstituted (*rac*)-ketone supplemented with NADH and glucose-6-phosphate dehydrogenase plus glucose-6-phosphate as a suitable coupled-enzyme cofactor recycling system. Both options were tested at a 100 mL scale in either an aqueous or 9:1 water–heptane reaction mixture after adjustment to pH 7.5. For the biooxidation runs, the added reactants were 1.4 g (*rac*)-alcohol plus 20 mg NAD^+^, whereas for the bioreduction runs, the corresponding additions were 0.4 g (*rac*)-ketone plus 20 mg NADH. The test systems were run at 20 °C for up to 72 h (water–heptane), or 48 h (aqueous). With hindsight, the choice to run both the biooxidative and bioreductive reactions at pH 7.5 was somewhat surprising, as by then, established studies with a number of other ADHs had shown that these two redox reactions have different pH-dependent activity profiles [53,54]. Nontheless, the reported outcomes from both unoptimised systems (Table 9) were impressive. Biooxidation of the (*rac*)-alcohol by BSADH in both the aqueous and water–heptane systems was characterised by a kinetic resolution that generated the corresponding (-)-(*1S*,*5R*)-ketone and (-)-6-*endo*-(*1R*,*5S*,*6R*)-alcohol. While the yields of the chiral ketone in the 24 h and 48 h aqueous systems were consistently higher than in the 48 h water–heptane system, the highest enantiomeric purity was recorded with the 48 h water–heptane system. Conversely, bioreduction of the (*rac*)-ketone by BSADH in both tested systems was also characterised by a kinetic resolution, in this case generating the corresponding (+)-6-*endo*-(*1S*,*5R*,*6S*)-alcohol and (+)-(*1R*,*5S*)-ketone. Complementary to the outcomes from the biooxidation runs, while the yield of the chiral alcohol in the 36 h aqueous systems was higher than in the corresponding 72 h water–heptane system, the higher enantiomeric purity was recorded with the water–heptane system. It was further reported that the activity of the aliquots of BSADH extract reclaimed from the various stopped reaction mixtures was approximately 70% of the originally prepared cell-free extract. Despite these various interesting outcomes and the long-established value of both chiral bicyclo[3.2.0]heptenones and corresponding heptenols as synthons for further chemocatalytic elaboration [82,83,108,109,110], no further developments of this intriguing BSADH system have ever been reported in the scientific literature.

After nearly two decades of investigations, commencing in 1979 with the Glaxo Group Research studies, it was apparent to the researchers at the University of Ferrara that no efficient biocatalytic means of generating enantiomerically pure 6-*exo*-bicyclo-[3.2.0]alkenols from the corresponding (*rac*)-alkenones had yet been reported. The exclusive bioreduction of the (*rac*)-ketones to the corresponding chiral 6-*endo*-alcohols had been the most frequently reported outcome, and somewhat ironically, the best enantiocomplementary redox result achieved (<18% yield, and e.e. 84% for the 6-*exo*-alcohol) had resulted from the very first such investigation by Newton et al. [81]. In an attempt to resolve this deficiency, the joint team from Ferrara and Bologna reverted to basics and instigated an extensive programme to screen a range of thus far untested yeasts and fungi with (*rac*)-bicyclo[3.2.0]hept-2-en-6-one and the corresponding 1-methyl-, 4-methyl-, and 1,4-dimethyl-substituted ketones [127]. The reported outcomes (Table 10) are notable in a number of respects. Somewhat disappointingly, with the exception of the bioreduction of the (*rac*)-1,4-dimethyl-substituted ketone by both *S. cerevisiae* RM74 and a *Trichoderma* sp., the consistent pattern of bioreduction of each of the four screened (*rac*)-ketones by the tested microorganisms resulted in the generation of predominantly corresponding *endo*-alcohols. However, there were some significantly differences in both the total yields and different relative amounts (and hence e.e.% values) of the corresponding (-)- and (+)-enantiomers. Interesting in this respect is that while *S. cerevisiae* RM9, like the commercial bakers’ yeast strain studied previously by Glaxo Group Research, bioreduced the unsubstituted (*rac*)-bicyclic ketone to principally the (+)-(*1S*,*5R*,*6S*)-*endo*-alcohol, two of the other tested yeast strains (RM1 and ML31) produced the corresponding (-)-(*1R*,*5S*,*6R*) enantiomer as the principal bioreduction product. The only other similar exception was an unclassified species of *Fusarium*, which produced the (-)-(*1S*,*5R*,*6R*)-*endo*-alcohol in low enantiomeric excess as the principal bioreduction product from the (*rac*)-1,4-dimethyl-substituted ketone substrate, a feature that distinguished it from the other three tested strains. With respect to 6-*exo*-alcohol production, the main objective of the screen, *S. cerevisiae* RM74 was the exception in producing enantiomerically pure (+)-(*1S*,*5R*,*6S*)-*exo*-alcohol as the major bioreduction product from the corresponding (*rac*)-1,4-dimethyl-substituted ketone, the only tested substrate. Conversely, *S. cerevisiae* RM1 failed to generate any bioreduced 6-*exo*-alcohol from the (*rac*)-4-methyl-substituted ketone. The remaining microorganisms all converted each of the tested (*rac*)-ketones to corresponding *exo*-alcohols as minor bioreduction products. However, as with the equivalent *endo*-alcohol products, there were some significant differences in both the total yields and different relative amounts of the corresponding (-)- and (+)-6*-exo*-enantiomers, resulting in most cases in the accumulation of a low level of single enantiomerically pure *exo*-alcohol.

In terms of the declared main objective of the 1996 screening exercise, finding the idiosyncratic pattern of bioreduction by *S. cerevisiae* RM74 could be viewed as the major successful outcome, a result that was confirmed on scaling up with 0.2 g of the disubstituted ketone introduced into 160 mL of a growing culture. However, based on the published scientific literature, no subsequent attempts were made to capitalise on this innovative result, or this atypical microorganism, not even by the team at Ferrara. Then, after the passage of more than a decade, another Italian multi-centre collaborative research team reintroduced a revised screening programme for relevant bioreductive activity in endophytic fungi sourced from plants growing in various locations, including the Amazonian forest of Ecuador [128]. The isolated microorganisms were tested for the ability to generate homochiral secondary alcohols by the single-step bioreduction of various ketones, including (*rac*)-bicyclo[3.2.0]hept-2-en-6-one. The four fungal isolates that bioreduced the bicyclic ketone most effectively all generated mixtures of the corresponding *endo*- and *exo*-alcohols, although to different relative extents, and with different degrees of stereospecificity (Table 11). Interestingly, in each case, the recovered (*1R*,*5S*,*6S*)-*exo*-alcohols were homochiral, as additionally was the (*1S*,*5R*,*6S*)-*endo*-alcohol generated by *Phomopsis* FE86, an endophytic fungus isolated from leaves of *Echinus molle* sourced from the forest at Quinto (Ecuador). No potential value for these homochiral bicyclic alcohols as synthons for subsequent chemocatalytic synthesis was commented on by the Italian team. While other subsequent screens have focused on endophytic biocatalysts [129], none to date has reported any corresponding activity with either unsubstituted or substituted bicyclo[3.2.0]carbocyclic molecules.

Considered as a whole, throughout the 30+ years of studying the ADH-dependent biocatalysis of bicyclo[3.2.0]carbocyclic molecules at both Ferrara and Glaxo Group Research, a consistant theme was in vivo studies that capitalised on the native enzyme complements of various culture collection strains. Less frequently used was in vitro screening using cell-free extracts or partially purified preparations replete with ADH activity. The latter clearly established that various HSDHs were the most successful isolated enzymes, possibly because of the fused polycyclic structure of their native steroid substrates. YADHs and HLADH produced less promising outcomes. Consistent throughout these in vitro studies has been a reliance on the biocatalytic activity of native enzymes. Despite the concurrent emergence of site-specific mutagenesis as a powerful genetic engineering technique to promote site-specific changes to enzymes, it has made little relevant impact on the biocatalysis of bicyclo[3.2.0]carbocyclic molecules by YADH, HLADH or HSDH [130]. The original methodology was introduced in the early 1980s [131,132,133], but was replaced by a much simpler protocol developed several years later by Kunkel [134], thereafter prompting studies on various ADHs. For YADH [135,136,137,138,139], the major interest has been to improve various aspects of performance, including those relevant to ethanol-based industrial processes. The interest in HLADH [140,141,142,143,144,145,146] has focussed on gaining a better understanding of the significance of the active-site architecture. Others less intensively studied ADHs include the corresponding enzymes from *Bacillus stearothermophilus* [147,148], *Clostridium autoethanogenum* [149], and *Pyrococcus furiosus* [150]. Additionally, extensive studies have been undertaken on representative HSDH enzymes from *Streptomyces hydrogenans* [151], *Brucella melitensis* [152], *Comomonas testosteroni* [153,154], *Arabidopsis* sp [155], as well as various human HSDH isoenzymes [156,157,158,159]. Despite these various endeavours undertaken with site-specific mutations of various ADHs and HSDHs, there remains a dearth of confirmed outcomes directly relevant to redox biocatalysis of bicyclo[3.2.0]carbocyclic ring molecules. While a study examining the effect on the biooxidation of various alcohols by The48Ser site-directed mutagenesis of human β-ADH confirmed a significant decrease in affinity for the polycyclic ketone 3β-hydroxy-5β-androstan-17-one (*K_m_* raised from 160 μM to 1 mM), no bicyclic secondary alcohols were tested [160]. However, other alternative initiatives involving redox-reversible ADHs acting in concert with redox-irreversible BVMOs to promote redox-neutral biocatalysis of bicyclo[3.2.0]carbocyclic molecules have generated significantly more positive outcomes, and the history of those advances will now be reviewed.

## 3. ADH-Dependent Redox-Neutral Biotransformations of Oxy-Functionalised Bicyclo[3.2.0]carbocyclic Molecules: The Evolution of Closed-Loop Artificial Biocatalytic Linear Cascades

As early as 1983, whilst still a member of Glaxo’s prostaglandin chemoenzymatic initiative, Stanley Roberts had recognised the cost implications associated with any relevant reliance on nicotinamide nucleotide coenzyme-dependent ADHs as stand-alone biocatalysts, and the potential solutions that could result from developing corresponding redox-neutral systems [89]. Although unable to actively develop this idea further whilst at Glaxo, circumstances changed when he subsequently left the company to take up a chair at the University of Exeter. This provided him with the opportunity to initiate and develop his own interdisciplinary research programme. thereby enabling him to pursue his interrelated interests in oxy-functionalised carbocyclic fused-ring systems, redox biocatalysis, and the chemoenzymatic synthesis of prostaglandins and other target molecules. Important to the evolution of the direction of this research initiative was his appreciation of the significance of a number of highly relevant prior publications. Formative in this respect were a series of reported studies by George Whitesides and Christopher Walsh. These included precedents for redox-neutral biocatalysis involving both NAD^+^/NADH [35], and NADP^+^/NADPH [37] coupled-enzyme cofactor recycling. Highly relevant to the biooxygenation of carbocyclic ring systems was their use of cyclohexanone monooxygenase (CHMO, EC 1.14.13.22), a 59-kDa monomeric Class A true flavoprotein sourced from *Acinetobacter* NCIB 9871 (now classified as *Acinetobacter calcoaceticus* NCIMB 9871). This enzyme is an NADPH-dependent Type I Baeyer–Villiger monooxygenase (BVMO [161]) originally recognised for its ability to biooxygenate cyclohexanone to ε-caprolactone, thereby serving as a key step in the pathway for the biodegradation of cyclohexanol by the bacterium [162]. Whitesides and Walsh recognised the potential of linking the activity of NADPH-dependent CHMO with that of NADP^+^-dependent glucose-6-phosphate dehydrogenase (G-6-PDH) isolated from *Leucanostoc mesenteroides* [163]. Significantly, the redox-neutral cooperative action of these two co-immobilised enzymes served to generate corresponding lactones from various unsubstituted and substituted mono-, bicyclo[2.2.1]- and tricyclo[3.3.1.1]alkanones. Given these outcomes, and Stanley Roberts’ own prior interest in bicyclic prostaglandin synthons, it was a novel extension of this lactone-generating coupled-enzyme technology that ultimately resulted in the success of his ‘Exeter BVMO-based lactone strategy’. Key to that development was the serendipitous emergence of the concept that combining the sequential biooxidative activities of an NAD(P)^+^-dependent ADH and an irreversible NAD(P)H-dependent BVMO with both complementary substrate and cofactor dependencies could in principle serve to promote a progressive alcohol → ketone → lactone(s) bioconversion without the need for an auxiliary cofactor recycling system (Figure 10A). Although not defined as such at the time, this innovative concept corresponds to a linear biocatalytic cascade with closed-loop cofactor recycling [164]. The unidirectional nature of BVMOs biocatalytic activity was a key operational feature of this closed-loop cofactor recycling system, which together with operating the initial ADH step at >pH 7.5 to favour biooxidative activity [53,54], should in principle result in complete conversion of an alcohol to its corresponding lactone products. This operational feature of ADH + BVMO closed-loop cofactor recycling distinguished it from the coupled-enzyme systems developed by Christian Wandrey [41,42] in which both complementary enzymes were reversible, resulting in compromised conversion rates with overall yields ≤ 50%, as reviewed by Hummel and Groger [165]. The current review will focus exclusively on the sequence of developments at Exeter that drove the evolution of combined ADH + BVMO redox-neutral closed-loop cascades. Other broader outcomes from the BVMO-based research programme developed at Exeter over the course of the decade 1989–1999 have been extensively reviewed previously [166], and subsequent iterative studies with this class of enzymes by other researchers comprehensively cross-referenced (vide infra).

It was the combination of his prior experience at Glaxo and an awareness of the precedents set by Walsh and Whiteside’s CHMO based published research [163] that defined from the outset Stanley Roberts’s choice of direction for the Exeter BVMO-based lactone strategy. That strategy commenced with the initial aim of generating valuable chiral synthons from carbocyclic bridged ring systems by deploying lactone-generating BVMOs as stand-alone biocatalysts, with the further aim of then progressing on to develop the corresponding but more challenging ADH plus BVMO closed-loop linear cascades. Because of the proven value of both bicyclo[3.2.0]-γ-lactones and bicyclo[2.2.1]-δ-lactones in Corey’s pioneering prior studies on the chemocatalytic synthesis of prostaglandins (Figure 3A [10]), the initial Exeter studies embraced both corresponding fused-ring ketone types as targeted moieties (Figure 10B). Commencing in 1987, three postgraduate research programmes (Helen Sandey, Andrew Carnell, Melissa Levitt) were actioned to investigate the potential value of BVMOs in single-step lactone-generating biocatalysis. Unlike in today’s post-genomic era, at the time that the Exeter studies were initiated, the choice of suitable enzymes that were readily available for delivering relevant lactone-generating biotransformations was strictly limited. Given the already established value of CHMO from NCIB 9871 [163], a survey of the then-published data base suggested that other potential candidate enzymes be included.

### 3.1. 2,5-Diketocamphane Monooxygenase (2,5-DKCMO)

This was the first ever BVMO confirmed to function as a cell-free activity [167]. It was isolated from (+)-camphor-grown *Pseudomonas putida* ATCC 17453 by Irwin Gunsalus’s research group at the University of Illinois [111,168,169,170,171,172]. It is a key enzyme in the degradation pathway for (+)-camphor, promoting the cleavage of the bicyclic terpene to an equivalent monocyclic pathway intermediate. Despite nearly a decade of significant and concerted investigation throughout the 1960s, much about the enzyme and its mode of action remained unclear, as evidenced by the frequently changing narrative used to describe the enzyme throughout that period [173]. It was, however, accepted that FMNH_2_ was the ultimate reductant essential to promote the lactonisation step [169]. The fact that a flavin moiety remained associated with the enzyme during purification led to the assumption that it was a true flavoprotein with an active site-bound FMN that sourced the protons to promote biooxygenation from a separate poorly characterised 36 kDa ‘NADH dehydrogenase’ [174]. This enzyme was not subject to SDS-PAGE analysis, but was subsequently purified and confirmed to be homodimeric (2 × 17.0 kDa [77]). Interest in 2,5-DKCMO and its perceived mode of action then declined, and so it was Gunsalus’s perspective that was taken as gospel in initially developing the Exeter BVMO-based lactone strategy. Subsequently, as a result of its FMN + NADH dependency, confirmed as an integral part of the Exeter studies [77], it was then classified as a Type II BVMO to distinguish it from the corresponding FAD + NADPH-dependent monooxygenases such as CHMO, which were conversely termed Type I BVMOs [161]. In subsequent post-Exeter studies undertaken nearly two decades later, Iwaki et al. [175] recognised that 2,5-DKCMO was in fact a flavin-dependent two-component monooxygenase (fd-TCMOs [176]), dependent on a separate NADH-dependent flavin reductase (FR) as the ultimate supplier of the requisite FMNH_2_,with the reduced flavin serving as a diffusible cosubstrate rather than an oxygenase-bound coenzyme. A camphor-induced dimeric FR (2 × 18 kDa) identified by Iwaki et al. that they termed ‘Fred’ was similar in many respects to the 36 kDa NADH dehydrogenase (2 × 17.0 kDa [77]) reported from Gunsalus’s 1960s studies with 2,5-DKCMO [174]. In one final twist to the saga, a relevant time-course study subsequently confirmed [177] that both Fred and the NADH dehydrogenase were in fact inducible enzymes associated only with secondary metabolism in ATCC 17453. Consequently, any in vivo functional role would be confined to post-exponential growth by ATCC 17453, and on further investigation corresponding roles were then confirmed for Frp1 and Frp2, two constitutive FRs present throughout exponential growth the bacterium [178].

### 3.2. 2-Oxo-Δ^3^-4,5,5-trimethylcyclopentenylacetyl-CoA 1,2-monooxygenase

This NADPH-dependent enzyme was also isolated from (+)-camphor-grown *P. putida* ATCC 17453 by Gunsalus’s research group at Illinois in the 1960s [179,180,181]. Although less well studied at the time than 2,5-DKCMO, it was confirmed to be an NADPH-dependent monooxygenase. It was presumed to be a BVMO responsible for ensuring ring-cleavage of the monocyclic pathway intermediate resulting indirectly from diketocamphane biooxygenation by 2,5-DKCMO [182]. It was then subsequently highly purified and extensively characterised in the early 1980s and confirmed to be a true flavoprotein with FAD serving as an active site-bound prosthetic group [183]. Its confirmed FAD + NADPH-dependency has resulted in it currently being classified, like CHMO, as a Type I BVMO [161].

### 3.3. Cyclopentanone Monooxygenase (CPMO)

This NADPH-dependent enzyme [184,185], induced by the growth of *Pseudomonas* NCIB 9872 (now classified as *Comamonas* sp NCIMB 9872) in cyclopentanol-based minimal medium, was similar in many respects to CHMO from *Acinetobacter* NCIB 9871 [186]. It was confirmed to catalyse the ring-expanding lactonising step in the biodegradation pathway of the monocyclic secondary alcohol cyclopentanol. The enzyme was identified and purified by researchers at the University of Aberystwyth in the 1970s. Unlike 2,5-DKCMO, CPMO was extensively characterised and.confirmed to be a true flavoprotein replete with active sitebound FAD, and sourcing the requisit reducing power for biooxygenation from NADPH. Like 2-oxo-Δ^3^-4,5,5-trimethylcyclopentenylacetyl-CoA 1,2- monooxygenase, the FAD + NADPH-dependency of CPMO resulted in it being classified as a Type I BVMOs [174].

Although the seminal and more extensive Exeter redox-based research was developed subsequently with inducible BVMOs from *P. putida* ATCC 17453 and *Acinetobacter* NCIB 9871 (a Class II nosocomial pathogen), considerations at the time based principally on the then-available containment facilities, ease of handling, and the extent of the relevant prior knowledge base effectively directed the choice of the initial target BVMO to the inducible NADPH-dependent CPMO present in cyclopentanol-grown *Pseudomonas* NCIB 9872. Consequently, this in vivo biocatalyst was used to commence ‘range-finding’ studies for the Exeter BVMO-based biocatalysis initiative in the autumn of 1988. Similarly, pragmatic considerations based on the inherent level of complexity associated with cofactor recycling resulted in the initial screening trials with CPMO being conducted with washed cell suspensions of the cyclopentanol-grown bacterium. Norbornan-2-one was trialled as the initial test substrate because of its perceived structural similarity to intermediates of the established cyclopentanol degradation pathway [184]. By the spring of 1989, the first promising results had emerged [187] when it was confirmed that a washed cell suspension of cyclopentanol-grown NCIB 9872 biooxidised the racemic unsubstituted bicyclic ketone to a recovered 38:1 mixture of the corresponding 2-oxa- and 3-oxa-lactones (Figure 11), which compares with an equivalent 9:1 ratio resulting from the peracid-catalysed chemical oxidation of the same ketone. Although the chiral purity of the resultant lactones was not determined at the time, the outcome was considered highly favourable because the predominant bridgehead lactone was seen as a potential synthon for the further chemocatalytic synthesis of various target molecules, including both nucleoside and prostaglandin analogues. However, before the relevant Exeter programme with CPMO from NCIB 9872 progressed further, a highly relevant outcome from a different BVMO-dependent bicyclic ketone lactonisation study was published by the research group at the University of Marseilles led by Roland Furstoss [188]. They reported that a washed cell culture of cyclohexane-1,2-diol-grown *Acinetobacter* TD63, a bacterium known to contain a relevant induced CHMO, biooxygenated (*rac*)-bicyclo[3.2.0]hept-2-en-6-one in a highly regio- and enantioselective manner to the corresponding (*1S*,*5R*)-2-oxa- and (*1R*,*5S*)-3-oxa-lactones (Figure 12). Of particular significance was that the one-step access to the optically active (-)-2-oxa-lactone opened up options to various Corey-type γ-lactones that themselves could then serve as a pivotal synthon for the further chemocatalytic synthesis of prostanoids. The elegance of this particular outcome was sufficiently profound to trigger a change of focus of the on-going BVMO-based research at Exeter. Rather than investigate the TD63 enzyme, the Exeter team redirected their CPMO-based programme to additionally include the equivalent NADPH-dependent CHMO, known to be induced in cyclohexanol-grown *Acinetobacter* NCIB 9871 [186], this being an enzyme with a proven ability to lactonize cycloalkanones [163]. This development was facilitated by the commissioning of suitable purpose-built containment facilities by the University. Consequently, a comparative study was made of the recorded outcomes from presenting separate washed cell suspensions of cyclohexanol-grown NCIB 9871 (CHMO) and cyclopentanol-grown NCIB 9872 (CPMO) with the disubstituted norbornanone (*rac*)-5-acetoxy-7-fluorobicyclo[2.2.1]heptan-2-one [189]. Contrasting outcomes were recorded for the two different biocatalysts (Figure 13). The cell suspension of NCIB 9872 generated a low yield (5%), the equivalent (*rac*)-5-hydroxy-7-fluoroketone as the only recovered biotransformation products. However, the cell suspension of NCIB 9871 produced a significantly different result. About half of the (*rac*)-5-acetoxy-7-fluoro-substituted norbornanone was biotransformed to a 4:1 mixture of the corresponding 5-hydroxy- and 2-oxa-lactone products. Although both were recovered, no chiral analysis was undertaken to establish the enantioselectivity, if any, of either biotransformation.The equivalent peracid-catalysed chemical oxidation generated a 4:1 mixture of the 3-oxa- and 2-oxa-bicyclooctanones. The outcome recorded with NCIB 9871 promoted an equivalent challenge with (*rac*)-5-bromo-7-fluoro-substituted norbornanone [190] resulting in a biooxidative kinetic resolution that generated a 1:1 mixture (40%:40% yields) of the corresponding (-)-(*1S*,*5R*)-2-oxa-lactone (e.e. >95%), and the recovered (-)-(*1R*,*4S*)-ketone (e.e. >95%). The equivalent peracid-catalysed chemical oxidation generated a 3.5:1 mixture of the 3-oxa- and 2-oxa-bicyclooctanones. In the light of the contrasting outcomes with the substituted norbornanones, a direct comparison was then made between the two different washed cell suspensions with unsubstituted (*rac*)-bicyclo[3.2.0]hept-2-en-6-one, and the equivalent 7-*endo*-methyl- and 7,7-dimethyl-substituted ketones [191]. The outcomes (Figure 14) served to highlight further significant differences between these two NADPH-dependent biocatalytic activities. Apart from the similar outcomes recorded with the (*rac*)-7-*endo*-methyl-substituted ketone, the recovered and characterised products from the other tested ketones again suggested that the inducible CHMO from NCIB 9871 was likely to prove to be the more promising biocatalyst for delivering potentially useful chiral synthons. Consequently, to begin to test the limits of its potential, a washed cell suspension of NCIB 9871 was challenged with the more highly substituted 2-*exo*-bromo-3-*endo*-hydroxy-7,7-dimethylbicyclo[3.2.0]hept-2-en-6-one, which again yielded a further promising outcome for this biocatalyst (Figure 14). Considered in the round, the possibility was advanced [191] that the inconsistent pattern of outcomes recorded with the various biooxygenations catalysed by washed cell suspensions of NCIB 9871 may have resulted from there being more than one competent BVMO enzyme in the bacterium, despite comprehensive prior evidence to the contrary [186]. However, after recording very similar outcomes with all four ketones on repeating the same series of biotransformation with a fresh washed cell suspension of NCIB 9871, that suggested possibility was withdrawn [192] in preference for ‘an alternative scenario involving one enzyme [CHMO] with a complex active site’. This cemented the collective conclusion drawn from the tests conducted thus far that the CHMO present in NCIB 9871 would prove to be the more useful enzyme for biooxygenating bicyclic fused-ring ketones.

The next tranche of relevant Exeter research, commencing in early 1991, focussed exclusively on cyclohexanol-induced CHMO from NCIB 9871. It was these particular studies that proved crucial to the evolution of the novel concept of redox-neutral closed-loop artificial biocatalytic linear cascades [2]. Serendipity and scientific progress have a long track record as a successful partnership, and so it proved to be at with the relevant studies at Exeter. Although a cell-free preparation of CHMO of undisclosed purity had been used previously to biooxygenate carbocyclic ketones into lactones as proof of principle [161], this was not considered to represent a practical option for further development. Of particular relevance was that the prior protocol depended on either a stoichiometric amount of NADPH, or the deployment of a complementary coupled-enzyme cofactor recycling with glucose-6-phosphate dehydrogenase. Both of these options would introduce significant scale-up cost issues. Consequently, the Exeter team initiated a series of studies to investigate an alternative solution. This was based on the possibility of developing a new in vitro protocol dependent on using fractured cell preparations of cyclohexanol-grown NCIB 9871. In principle, this would both provide unfettered access of added ketone substrates to the NADPH-dependent CHMO, and drive any resulting lactonisations to completion by capitalising on residual cyclohexanol in the fractured cell preparation plus the concerted activity of NADP(H)-dependent cyclohexanol dehydrogenase (CHDH, EC 1.1.1.245), an effective cyclohexanol-induced secondary alcohol dehydrogenase induced by cyclohexanol in NCIB 9871 [186]. (*rac*)-5-Bromo-7-fluorobicyclo[2.2.1]heptan-2-one was chosen as the initial test substrate because it was known to give a characteristic outcome with cyclohexanol-grown NCIB 9871 (Figure 13 [189]. It was predicted that an initial closely monitored time-course study would confirm a direct one-step kinetic resolution progressing towards an equimolar mixture of the (-)-(*1R*,*4S*)ketone, and the corresponding (-)-(*1S*,*5R*)-2-oxa-lactone. However, when analysed, the resultant data proved to be anything but predictable—rather, it proved to be transformational for the future of the Exeter-based research initiative. GC analysis of the recovered samples taken at every 30-min interval throughout the first 2.5 h period of the study confirmed that the initial concentration of the introduced test (*rac*)-disubstituted-bicyclo[2.2.1]ketone progressively decreased by more than 75% (Table 12). However, rather than resulting in the anticipated progressive lactone-generating biooxygenation, the analysed samples demonstrated a concomitant progressive increase in the corresponding (*rac*)-*endo*-bicyclo[2.2.1]alcohol.

This unexpected outcome was interpreted as resulting from an initial non-enantiospecific bioreduction catalysed by NADP(H)-dependent CHDH, or possibly another previously unrecognised ADH enzyme(s) present in the fractured cell preparation of cyclohexanol-grown NCIB 9871. Analysis also confirmed that the detectible level of the corresponding 2-oxa-bicyclo[3.2.1]lactone in the reaction mixture sampled at 2.5 h was very low. Subsequent monitoring over the following 19.5 h period highlighted a number of consistent trends. Thus, the concentration of the (*rac*)-*endo*-bicyclo[2.2.1]alcohol progressively decreased, while that of the (*rac*)-bicyclo[2.2.1]ketone initially rose but subsequently declined, signalling the initiation of NADPH-dependent CHMO activity, as indicated by a concomitant progressive increase in the accumulation of the corresponding (-)-(*1S*,*5R*)-2-oxa-bicyclo[3.2.1]lactone. Analysis of duplicate stopped 22 h samples confirmed that both the (-)-ketone (40% estimated yield, e.e. >95%) and (-)-2-oxa lactone (40% estimated yield, e.e. >95%) were of high optical purity. An otherwise equivalent biotransformation undertaken with the addition of 2 μM rifampicin (a potent inhibitor of prokaryotic RNA polymerase [193]) resulted in an almost identical progressive sequence of outcomes [77]. This implied that the strictly sequential combination of the enzyme-catalysed biotransformations: NADPH-dependent CHDH (reductive: reaction 1), followed by NADP-dependent CHDH (oxidative: reaction 2), and subsequently NADPH-dependent CHMO (reaction 3) were catalysed by pre-induced rather than de novo enzyme activities in the fractured cell preparations:



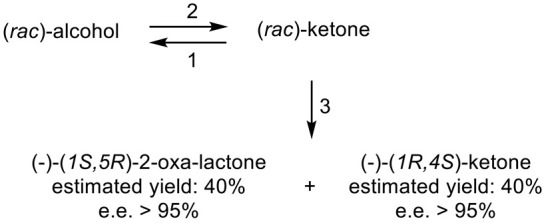



Although not specifically investigated, it was considered at the time that the initial K → A → K ‘push–pull’ redox exchange may have been pH-driven, as ADHs are redox-reversible enzymes dependent on the ionisation state of the participating amino acid residues, a known influential factor for ADHs [53,54]. More recent post-Exeter research, however, has suggested that the outcome may represent some form of thermodynamic stochastic fluctuation [194,195,196] to the NADP^+^/NADPH balance of the fractured cell suspensions promoted by the addition of the significant titre of the abiotic test substrate. A similar concept of ‘thermodynamic stabilisation’ has been proposed even more recently to characterise an equivalent ADH-dependent redox exchange [197].

It was only on reflection that this unexpected succession of redox biotransformations was seen as suggesting that some form of cofactor recycling system between CHDH and CHMO may have been operating to promote the biotransformation of the test substrate by the fractured cell preparation of NCIB 9871. Further evidence for this proposal was then gained from the subsequent study that confirmed that when a fractured cell sample on NCIB 9871 was challenged initially with the (*rac*)-5-bromo-7-fluorobicyclo[2.2.1]heptanol (Table 13), the outcome closely mirrored that recorded after the initial 2.5 h of the corresponding (*rac*)-ketone biotransformation.

This outcome provided the first concrete evidence for the involvement of a closed-loop cofactor recycling system. Because of the novelty of these outcomes, it was decided to closely monitor the biotransformation by fractured cell sample of cyclohexanol-grown NCIB 9871 of (*rac*)-bicyclo[3.2.0]hept-2-en-6-one because this abiotic ketone was known to be rapidly biotransformed by NADPH-dependent CHMO to yield a mixture of regioisomeric lactones (Figure 14 [191]). Again, the same successive, albeit more rapid, pattern of ketone →alcohol → ketone → lactone molecular interchanges occurred, resulting after 3 h in a 1:1 mixture of the corresponding 2-oxa- and 3-oxalactones being recovered from the reaction mixture. When challenged with the corresponding (*rac*)-6-*endo*-bicyclo[3.2.0]hept-2-en-6-ol, the simpler sequence of redox biotransformations resulted, characterised by an initial progressive direct biooxidation of the (*rac*)-alcohol to the corresponding (*rac*)-ketone, accompanied by a coincident, but slower, irreversible biooxygenation of the generated (*rac*)-ketone to a 2:1 mixture of the (*1S*,*5R*)-2-oxa-lactone (e.e. 86%) and (*1R*,*5S*)-3-oxa-lactone. Together, these outcomes with the (*rac*)-bicyclo[3.2.0]test substrates strongly supported the concept of closed-loop cofactor recycling.

Further strong evidence to support the conclusions drawn from the fractured cell studies came from a related study that then assayed the titres of both NADP(H)-dependent CHDH and NADPH-dependent CHMO activities in the cells of NCIB 9871 throughout the 8 h period immediately following inoculation of the nutrient agar-maintained bacterium into freash cyclohexanol-based minimal medium (Table 14 [198]). Both enzymes were confirmed to be inducible, but on different progressive time scales throughout the monitored time period, covering the successive lag and log phases of growth. This correlates with the established concept of ‘from the top coordinate induction’ for the progressive expression of the enzymes of many microbial catabolic pathways [199]. With the exception of late log phase growth, significant levels of activity of both NADP(H)-dependent CHDH and NADPH-dependent CHMO were recorded throughout the monitored time-window. Based simply on the relative levels of the two participating enzymes, the maximum potential for redox-neutral closed-loop cofactor recycling cooperation occurred somewhere between 2 and 6 h after inoculation under the growth conditions used. The consensus conclusion from these related studies with fractured cell preparations of NCIB 9871 was that the activities of a reversible secondary alcohol dehydrogenase (most probably NADP(H)-dependent CHDH) and an irreversible Baeyer–Villiger monooxygenase (most probably NADPH-dependent CHMO), both pre-induced by growth of the bacterium on cyclohexanol, were operating a form of closed-loop cofactor recycling that served to facilitate the indirect generation of lactone(s) from a corresponding alcohol via the intermediacy of the corresponding ketone (Figure 10A). Using more recently introduced terminology [162], this two-step bioconversion catalysed by fractured cells of NCIB 9871 corresponds to a redox-neutral natural in vitro biocatalytic linear cascade. The irreversible nature of the BVMO-catalysed reaction is an important feature, as it serves to drive the sequence to lactone production. In this respect, the pioneering closed-loop system offers a significant advantage compared to the coupled-enzyme systems developed previously by Wandrey et al. [41,42].

A logical next step was to determine whether the secondary alcohol-initiated two-step biotransformation could be conducted with equal ease with corresponding isolated NAD(P)H-dependent enzymes. The purified NADP(H)-dependent CHDH had been reported previously to be relatively unstable [186]. Consequently, TBADH was chosen for inclusion as the secondary alcohol dehydrogenase both because of its known robustness [200] and its relatively cheap cost. Taking into account the previously reported pH optima of both CHMO (7.1–7.3 [186]) and TBADH (7.8–8.2 [201]), plus the predisposition of ADHs to function biooxidatively at pH > 7.5 [53,54], the concept was tested initially in 40 mL of Tris-HCl buffer (0.1 M, pH 8.0) supplemented with 66 mg (*rac*)-6-*endo*-bicyclo[3.2.0]hept-2-en-6-ol, 40 U of purified CHMO [186], 100 U of a commercial preparation of TBADH (Sigma-Aldrich, St. Louis, MO, USA), and 150 mg of the sodium salt of NADP^+^ (Sigma-Aldrich) [2]. After 5.5 h in an orbital incubator at 30 °C, the outcome of this one-pot biotransformation was that the (*rac*)-alcohol had been biotransformed (79% conversion) to a 2:1 mixture (41% isolated yield) of the corresponding 2-oxa- and 3-oxa-lactones: fuller characterisation then focussed on the predominant bridgehead lactone which was confirmed as being the (-)- (*1S*,*5R*)-enantiomer (e.e. 86%). At the time, this result was described as representing ‘a coupled-enzyme system characterised by closed-loop cofactor recycling’ (Figure 15 [2]). Compared to the previously developed cell-based systems, this outcome represented an example of an artificial linear biocatalytic cascade [162], referred to as the in vitro redox-neutral ‘Exeter-I system’ [202].

Having developed to the proof-of-principle stage, both cell-based and isolated enzyme-based NADP(H)-dependent closed-loop biocatalytic linear cascades exploiting FAD + NADPH-dependent CHMO (a Type I BVMO [174]) able to deliver chiral regiospecific bicyclic lactones, attention then turned to developing equivalent in vitro artificial NAD(H)-dependent ADH + BVMO systems. The considerable relative cost differential between the two alternative systems (principally NADP(H) versus NAD(H)), especially when considered at any potential subsequent scale-up stage, was the principal driver for this initiative.

While sourcing known potentially suitable NAD(H)-dependent ADHs such as YADH, HLADH, and HSDH was relatively straightforward, conversely the choice of then-available Type II FMN + NADH-dependent BVMOs was extremely limited [161]. The NADH-dependent luciferases found in various bioluminescent bacteria were discarded as a potential option because of their known limited chemocatalytic specificity [203], most likely resulting from their evolved niche role in nature. This left the known isoenzymic DKCMOs induced in camphor-grown *P. putida* ATCC 17453 [172] as the only potential viable options. While both 2,5- and 3,6-DKCMO were known to generate the same pathway intermediate (2-oxo-Δ^3^-4,5,5-trimethylcyclopentenylacetic acid [OTE]) from 2,5- and 3,6-diketocamphane, respectively [168], it was also known that the further catabolism of OTE by camphor-grown ATCC 17453 involved 2-oxo-Δ^3^-4,5,5-trimethylcyclopentenylacetyl-CoA 1,2-monooxygenase, a Type I NADPH-dependent BVMO [183]. This combination of Type I and Type II BVMOs in the same microorganisms effectively ruled out meaningful in vivo studies with whole or fractured cell preparations of ATCC 17453. Learning how to capitalize on the DKCMO activity proved challenging at times, principally because at the time (mid-1993), the isoenzymes were both incompletely and incorrectly characterised, and it would be two decades later before their correct status as flavin-dependent two-componenent monooxygenases (fd-TCMOs) was established (vide supra [175,177]). As a consequence, the isoenzymic DKCMO-based programme of bicyclic ring system biocatalysis at Exeter was characterised by a steep learning curve. Retrospectively, a number of the early outcomes of that initiative are better understood with an appreciation of some relevant idiosyncratic characteristics of ATCC 17453 and the isoenzymic DKCMOs that evolved both as a direct result of contemporaneous Exeter studies and subsequently [175,176,177,178]. The most salient features of the DKCMO-based contribution to the ‘Exeter BVMO-based lactone strategy’ were:MO1 was used for a number of the trialled biotransformations. This was a partly purified preparation (60–75% saturated (NH_4_)_2_SO_4_ ‘cut’) from a cell-free extract of either (+)- or (-)-camphor-grown ATCC 17453 harvested at the mid-exponential phase of growth. Growth on either medium induced equivalent highly active titres of both isoenzymic DKCMOs. No detectible titres of either 2-oxo-Δ^3^-4,5,5-trimethylcyclopentenylacetyl-CoA 1,2-monooxygenase or a secondary alcohol dehydrogenase activity relevant to either bicyclic alken- or alkanols was detectible in MO1. Specifically relevant in this latter respect was the absence of any titre of camphor-induced NAD(H)-dependent *exo*-hydroxycamphor dehydrogenase (EC 1.1.1.327 [204,205]). Although mid-exponential phase cells were used to prepare MO1, it contained extremely low titres of both Fred [175] and the NADH dehydrogenase [168,170,172], but did contain significant titres of Frp1 and Frp2, both being NADH-dependent FRs [175,176,177,178,206,207,208].Purified 2,5-DKCMO and 3,6-DKCMO were prepared from harvested cells of ATCC 17453 grown in either (+)-camphor- or (-)-camphor-based media. Both purified enzyme preparations contained no detectible titres of 2-oxo-Δ^3^-4,5,5-trimethylcyclopentenylacetyl-CoA 1,2-monooxygenase, or any secondary alcohol dehydrogenase activity relevant to either bicyclic alken- or alkanols. While both preparations contained no detectible titre of Fred, and only a barely detectable titre of the NADH dehydrogenase, they did retain significant detectable titres of both Frp1 and Frp2 [206]. This enabled purified preparations of the DKCMO to biooxygenate abiotic ketones when co-presented with NADH [172,207,208].Whereas the repressor protein(s) controlling transcription of the relevant DKCMO genes in ATCC 17453 [175] were cathodic in specificity as evidenced by the outcomes reported in 1 and 2 above, the corresponding purified DKCMO isoenzyme displayed absolute chemo-, regio- and stereospecificity towards the corresponding enantiomer of camphor when used as a test ketone substrate. Contrastingly however, they exhibited directly equivalent patterns of selectivity when challenged with a number of abiotic *(rac*)-bicyclo[3.2.0]- and (*rac*)-bicyclo[2.2.1]ketones [161,166]. The outcomes from these tested bicyclic ketones consistently confirmed that 2,5-DKCMO was the more specific isoenzyme [173,207,208].

Relevant DKCMO-based research at Exeter directed towards developing an ADH + BVMO NAD(H)-dependent in vitro artificial biocatalytic cascade commenced in 1993. Because of its relative ease of preparation, MO1 was considered a suitable biocatalyst for trial studies because it was known to contain high titres of both DKCMO isoenzymes, low titres of Fred and NADH dehydrogenase, and no detectable titres of either 2-oxo-Δ^3^-4,5,5-trimethylcyclopentenylacetyl-CoA 1,2-monooxygenase or native secondary alcohol dehydrogenase(s). The initial trial run was based on the lessons learnt from the original successful artificial linear TBADH-CHMO-dependent Exeter-1 system. The test substrate used was (*rac*)-6-*endo*-bicyclo[3.2.0]hept-2-en-6-ol, but in this case the artificial coupled-enzyme system comprised MO1, added NAD^+^, and a commercial preparation of the NAD(H)-dependent HLADH, a secondary alcohol dehydrogenase with very broad chemospecificity, but little or no enantiospecificity [209]. The biotransformation was monitored by GC, confirming that no detectible bicyclic alcohol or ketone remained after 60 min. Conversely, analysis confirmed that equivalent high yields of both the corresponding (*1R*,*5S*)-2-oxa- and (*1S*,*5R*)-3-oxa-lactones had been produced, and in each case by equivalent highly regio- and enantiospecific biooxygenations. Collectively, these outcomes confirmed the first successful example of an ADH + Type II BVMO-dependent artificial closed-loop biocatalytic linear cascade, referred to as the in vitro redox-neutral Exeter-II system (Figure 16 [202]). Also highly relevant was that this result confirmed that MO1 exhibited enantiocomplementary specificity to that of the CHMOs from both *Acinetobacter* TD63 [188], and *Acinetobacter* NCIB 9871 [189,190]. This was considered a highly significant outcome as it signalled the possibility that the choice of biocatalyst could provide access to synthons in enantiocomplementary series. However, it was not possible to establish from this isolated result whether only one of the isoenzymic DKCMOs present in MO1 was active with this ketone, or whether both isoenzymes were active with very similar regio- and enantiospecificities. Given this promising initial outcome with MO1, it was clearly important to resolve the relative contributions made by 2,5-DKCMO and 3,6-DKCMO. It was considered that the most direct way to do this was by challenging purified preparations of each isoenzyme separately with (*rac*)-bicyclo[3.2.0]hept-2-en-6-one plus NADH. Analysis of the 60 min stopped reaction mixtures (Table 15 [202]) confirmed firstly that 2,5-DKCMO was the more active isoenzyme in terms of total lactones formed. While they both biooxygenated the test ketone with the same regiospecificity, they did so with significantly different degrees of the same enantiospecificity. The 2,5-DKCMO isoenzyme clearly introduced a greater degree of chirality with respect to the generated lactones in both regioisomeric series. Based on the then-current knowledge of the participating enzymes, it was assumed, based on the proposals made by Gunsalus et al. in the mid-1960s [210,211], that each isoenzyme was deploying the albeit low titre of NADH dehydrogenase to generate the FMNH_2_ necessary for DKCMO-dependent lactonisations (Figure 17) from oxygenase-bound FMN serving as a coenzyme. However, given the confirmed presence of significant titres of Frp1 and Frp2 in both purified isoenzyme preparations, and the now-known role of these two flavoprotein reductases in generating FMNH_2_ to serve as a cosubstrate [173,177,178], this suggests that an alternative linked enzyme system may have been responsible for the results recorded (Table 15) with the purified DKCMO isoenzyme preparations (Figure 17). By extension, the prior outcome recorded with MO1, HLADH, (*rac*)-alcohol and NAD^+^ could be explained by the HLADH plus the (*rac*)-alcohol serving as a source of NADH for Frp1/Frp2 to then drive a closed-loop redox-neutral biotransformation.

Subsequently, studies at Exeter were directed towards expanding the potential of the DKCMOs from ATCC 17453 for undertaking practical biotransformations. The novelty of the redox-neutral ADH + BVMO Exeter-II system (Figure 16 [202]), offering efficient NAD(H) closed-loop cofactor recycling proved a highly tempting attraction, so an initial replicate study was undertaken [212] that convincingly confirmed the earlier promising outcome. Attention was directed to a proof of principle study that would incorporate a compatible initial hydrolytic step able to yield a corresponding secondary alcohol, and thereby establish a linear three-enzyme cascade. Given the then current extensive interest in using various lipases as biocatalysts [213,214], a relevant study was undertaken with the oxy-functionalised bicyclic ester (*rac*)-*endo*-6-acetoxybicyclo[3.2.0]hept-2-ene as a test substrate. Included in the one-pot reaction mixture were NAD^+^, MO1, HLADH, and a commercial preparation of *Pseudomonas cepacia* lipase [212]. The flask was incubated at 30 °C in an orbital shaker for 120 min, and samples removed for analysis by GC at 30 min intervals. The time-course data (Table 16) indicated an initial lipase-catalysed hydrolysis of the *endo*-acetate resulting in a significant decrease by 45% after only 30 min. Similar levels of the corresponding *endo*-alcohol (28%) and ketone (19%) were detected in the initial 30 min sample, but both progressively declined thereafter to barely detectable levels by 120 min. Concomitantly, from a combined low level (8%) when sampled at 30 min, the level of the corresponding regioisomeric 2-oxa- and 3-oxa-lactones then progressively increased to 9% and 32%, respectively, at 120 min. Subsequent chiral analysis of the stopped 120 min mixture confirmed that the only three recoverable products were the (*1S*,*5R*,*6S*)-*endo*-ester (e.e. >95%), the (*1S*,*5R*)-2-oxa-lactone (e.e.% >93%) and the (*1S*,*5R*)-3-oxa-lactone (e.e. >98%).

Overall, these outcomes were described [212] as the lipase, HLADH, and MO1 cooperatively acting as an artificial biocatalytic linear cascade incorporating closed-loop cofactor recycling of NAD(H) (Figure 18). The one-pot three-enzyme sequence initially performed a kinetic resolution of the (*rac*)-acetoxy ester to yield the (*1R*,*5S*,*6R*)-*endo*-alcohol: subsequently, the chiral alcohol was firstly biooxidised and then biooxygenated by HLADH and MO1, respectively, to yield both corresponding chiral regioisomeric lactones as recovered end products. Taking into account the established temperature optima for *P. cepacia* lipase (45–50 °C [215]), HLADH (38–43 °C [216]), and both the DKCMO isoenzymes (35–38 °C [165]), an equivalent multi-enzyme biotransformation was then run at 38 °C. This resulted in the same recorded sequential pattern of enantiospecific outcomes, but which ran more rapidly to completion by 90 min.

Having successfully confirmed at proof-of-principle stage both two-step and three-step artificial biocatalytic linear cascades of bicyclic molecules deploying closed-loop cofactor-recycling, potential scale-up considerations assumed significance. Capitalising on the personal friendship between Stanley Roberts and Giacomo Carrea at Milan University, an initial relevant investigation was aimed at increasing the efficiency of closed-loop cofactor recycling. Based on Wandrey et al.’s methods to produce L-amino acids from the corresponding racemic α-hydroxy acids [41,42], Carrea had developed the equivalent use of polyethylene glycol (PEG)-immobilised preparations of NADP(H) in membrane reactors to stabilise the lactonising activity of NADPH-dependent CHMO for several days [217]. An equivalent PEG-NAD^+^ covalently attached preparation was prepared for use in a small bioreactor comprising a 10 mL ultrafiltration unit fitted with MW 3000 kDa cutoff membranes. At this preliminary experimental stage, it was decided to test the system initially with a comparatively simple coupled-enzyme system deploying MO1 plus FDH to biotransform (*rac*)-bicyclo[3.2.0]hept-2-en-6-one [218]. The bioreactor was charged with 0.05M phosphate buffer, 0.05 M sodium formate, 0.5 mM dithiothreitol, 30 mM (*rac*)-ketone, 10 U of commercial FDH, and 50 mg of lyophilised MO1 (Figure 19). The combined activities of FDH plus the Frp1/Frp2 and isoenzymic DKCMO titres present in MO1 served as a three-step cofactor recycling system. The charged bioreactor was incubated at 30 °C for 24 h. The low-MW spent reaction mixture was then discharged from the mem-brane reactor for analysis, and replaced with an aliquot of fresh reaction mixture for incubation for a further 24 h. This discharge–recharge cycling was repeated for a further 5 days, at which point the membrane reactor was replenished with fresh MO1, FDH and reaction mixture. Following this comprehensive recharge, the membrane reactor was run deploying 24 h charge–discharge cycles for a further 120 h. The resulting data when fully analysed (Table 17) were highly promising, confirming that the DKCMOs in MO1 had operated continuously for 12 days to deliver both 2-oxa- and 3-oxa-lactones with degrees of regio- and enantiospecifiity that matched those recorded previously with freshly prepared partially purified MO1.

An additional incentive was provided by fresh funding from Chiroscience Ltd. to develop an equivalent bioreactor able to biooxygenate 2-(2′-acetoxyethyl) cyclohexanone to the corresponding chiral 2-oxa-lactone, a key synthon for the subsequent chemocatalytic synthesis of the valuable antioxidant (*R*)-(+)-lipoic acid. An initial range-finding one-pot trial with the proven MO1 + FDH coupled-enzyme system resulted a very low yield (8%) of the (*R*)-(-)-2-oxalactone. This outcome then prompted a screen for a more efficient monooxygenase, resulting in the recognition of NADPH-dependent 2-oxo-Δ^3^-4,5,5-trimethylcyclopentenylacetyl-CoA 1,2-monooxygenase from camphor-grown *P. putida* ATCC 17453 as a promising alternative biocatalyst [219]. Like MO1, it was used as a semi-purified preparation, but termed MO2. When then tested with glucose-6-phosphate dehydrogenase plus glucose-6-phosphate as the coupling enzyme system, MO2 yielded 36% of the (*R*)-(-)-lactone in high enantiomeric excess (83%).

The intention was then to investigate further the potential of both the NADH-dependent and NADPH-dependent systems, either as immobilised cofactor- or co-immobilised enzyme-based bioreactor systems, able to deliver valuable synthons. Together with expanding the Exeter-I and Exeter-II redox-neutral cofactor recycling initiatives, this would have generated a strong comprehensive nucleus of redox research at Exeter. That was a challenging vision that would have required a considerable investment in additional infrastructure and financial support to be provided by the University. However, neither materialised, and consequently Stanley Roberts and a number of the Exeter interdisciplinary biocatalysis team subsequently reinvented themselves elsewhere.

In retrospect, a major legacy of the Exeter BVMO-based strategy can be seen as the introduction of both in vivo native and in vitro artificial biocatalytic linear cascades deploying closed-loop cofactor recycling, thereby eliminating the necessity for an auxillary cofactor recycling system: the irreversible nature of the terminal enzymes deployed was also strategically important. A corollary was that the strategy delivered one-pot one-substrate biocatalytic cascades precluding co-product formation with consequential expensive downstream processing implications. Additionally important from the perspective of chemoenzymatic synthesis was those routes delivered enentiocomplementary homochiral lactone synthons. Following the cessation of the Exeter initiative, wider interest in the redox-neutral closed-loop cofactor recycling systems remained dormant for a number of years. However, understanding and exploiting the value of BVMOs to catalyse useful biooxygenations was subsequently taken on and expanded extensively at other research centres. A principal driver for a number of these developments was the growing influence of ever-more-powerful genetic engineering techniques on biocatalysis. Typically, following the pioneering research of Frances Arnold [220], directed evolution and rational design enabled significant changes to be made to native enzyme chemo-, regio- or stereospecificity [221,222,223,224,225]. This proved an attractive option for altering the catalytic characteristics of various BVMOs [226,227,228,229,230,231,232,233], and a number of the resultant studies included a single test assay conducted with (*rac*)-bicyclo[3.2.0]hept-2-en-6-one as a ‘model substrate’ [234,235,236,237,238,239,240]. This enabled direct comparability with the accepted bench-mark outcome for this (*rac*)-ketone recorded previously with the CHMO from *Acinetobacter* TD63 [188], and the DKCMOs from *P. putida* ATCC 17453 [198]. Genome mining for novel predicted BVMO activities was another valuable development to emerge from the postgenonomic era, resulting in the subsequent isolation of a significant number of previously untested enzymes [241,242,243,244,245]. The extent of the influential effect of these genetic engineering techniques is evidenced by the sheer number of relevant papers published since the start of the current century. Although there were some notable in vivo-based studies, including the scale-up to 50 L of the lactonisation of (*rac*)-bicyclo[3.2.0]hept-2-en-6-one by a recombinant *E. coli* strain overexpressing the CHMO from NCIB 9871 [246,247], most attention was focussed on in vitro studies of NADPH + FAD-dependent Type I enzymes. While using isolated BVMO enzymes can be advantageous for a number of reasons, this did impose a requirement some form of relevant cofactor recycling. The most favoured options were various conventional coupled-enzyme systems (vide supra).

Eventually, after over a decade, interest in the pioneering NADP(H)-dependent Exeter-1, and NAD(H)-dependent Exeter-II redox-neutral ADH + BVMO closed-loop systems was rekindled in the wake of a wider interest in cascade biocatalysis in general [153]. A succession of publications commencing at the beginning of this century [248,249,250,251,252,253,254,255] served to reinforce some of the outcomes established by both the initial in vivo, and later in vitro Exeter studies. While the potential value of conducting in vivo multi-step biocatalytic procedures using whole cells as a means to preclude external cofactor recycling was recognised [256,257,258,259,260,261], it was the advantages offered by the in vitro one pot artificial biocatalytic linear cascades [202,212] that attracted more attention. This resulted from both the important strategic value of avoiding both time-consuming and/or yield-reducing isolation and purification steps for relevant intermediates, and the particular merits of the closed-loop cofactor recycling methodologies. First to be developed were various in vitro redox-neutral closed-loop cofactor recycling systems involving exclusively stereocomplimentary pairs of ADHs operating redox-neutral closed-loop cofactor recycling to facilitate the biocatalytic racemisation of pairs of either secondary alcohols [262,263] or α-hydroxycarboxylic acids (Figure 20 [264]). Another version of this type of linear cascade used redox-neutral closed-loop cofactor recycling to drive the deracemisation of mandelic acid to phenylglycine by the successive activity of D-mandelate dehydrogenase and a commercial reversible amino acid dehydrogenase (Figure 20 [265]. However, all these systems were influenced by the same thermodynamic stochastic fluctuation [194,195,196,197] encountered by Wandrey when developing equivalent redox-neutral systems in the 1980s [41,42]. Similar ADH-triggered redox-neutral cascades were then developed by Tong et al. (Figure 21 [266]), and Gargiulo et al. (Figure 21 [267]), although the latter was notable for deploying an irreversible enoate reductase as the terminal biocatalyst which served to eliminate thermodynamic ‘push–pull’ fluctuation [194,195,196,197]. An interesting development was the combination of an initiating alkanol-generating monooxygenase (CYP BM3 from *Bacillus megaterium*) plus a subsequent complementary ADH to function as an alkanone-generating redox-neutral cofactor recycling system to biotransform n-alkanes [268] and cycloalkanes (Figure 22 [269]). Significantly, because the reversible oxidoreductase served as the terminal biocatalyst, again the efficiency of these systems was subject to influence by thermodynamic fluctuation [194,195,196,197]. In turn, this disadvantage could be overcome by introducing as a terminal step an additional BVMO-catalysed step so generating a lactone-forming three-enzyme cascade [270]. While none of these linear biocatalytic cascades corresponded to either the redox-neutral Exeter-I or Exeter-II ADH + BVMO-dependent thermodynamically driven closed-loop cofactor recycling systems developed in the 1990s, that all changed in 2013, when, acknowledging the pioneering Exeter studies, Uwe Bornschueur’s group at Greifswald University recognised that substituting cyclohexanol for bicyclo[3.2.0]heptanol as the initiating substrate should in principle generate ε-caprolactone, a versatile commodity chemical because of its direct and indirect roles as a monomer for various useful polymers [271]. Consequently, Greifswald developed and reported what they termed “a self-sufficient Baeyer-Villiger biocatalysis system for the synthesis of ε-caprolactone from cyclohexanol” (Figure 23 [272,273]), which corresponded to the original Exeter-II closed-loop system [202,212]. The Greifswald system did differ not only in the feedstock alcohol, but in also in depending on genetically modified rather than native enzymes. The polyol dehydrogenase (PDH) gene from *Deinococcus geothermalis* was genetically modified for better relevant substrate and cofactor acceptance and overexpressed in *E. coli*, as was the CHMO from NCIB 9871. To facilitate enzyme reuse and enhance stability, the initial results gained with soluble preparations of the enzymes tested separately were used to progress to a system deploying co-immobilised PDH and CHMO using the RelizymeTM HA403 support matrix. Operating as a redox-neutral in vitro biocatalytic linear cascade with closed-loop NADP(H) cofactor recycling to transform cyclohexanol (5 mM) to ε-caprolactone, the system was monitored through five successive 1-hour cycles, with a biocatalyst recovery, washing, and recharging step included between each cycle (Table 18). However, the reported performance of the system was disappointing when compared to that of the previously developed Exeter membrane reactor (Table 17 [218]), albeit the latter in vitro system was run with immobilised cofactor and soluble enzymes rather than co-immobilised enzymes. Other more complex redox-neutral in vitro linear cascades were also developed for the stereoinversion and deracemisation of secondary alcohols (Figure 22 [274]), but these depended on two discrete consecutive oxidation and reduction cycles, each deploying completely separate NAD(H) and NADP(H) cofactor recycling systems, and each consequently requiring a separate sacrificial cosubstrate. However, like the parallel interconnected kinetic asymmetric transformation system (PIKAT) proposed by Vicente Gotor [275,276], these were not closed-loop systems; rather, they represented a more complex version of the original coupled-enzyme concept first reported by Levy et al. over 50 years previously [31]. Approached from a different perspective, an interesting in vitro cofactor recycling concept that offered an alternative strategy to the Exeter-developed closed-loop cascade system to promote effective cofactor recycling was the development of ADH-BVMO fusion enzymes [277,278,279,280]. However, as with many aspects of genetic engineering, theory is not reality, and consequently success can prove to be elusive. Careful design of such chimeric bifunctional enzymes proved to be challenging in order to both optimise stability and promote effective substrate and cofactor channeling within the chimeric complexes. These developments have been reviewed comprehensively in context by Aalbers and Fraaije [281], and have continued to be progressively updated [282].

Also recognised was the potential advantage to be gained from an in vivo approach by engineering desired multi-step procedures in whole cells, an approach sometimes referred to as ‘systems biocatalysis’, thereby eliminating the dependence on external cofactor recycling [256,257,258,259,260,261]. Successfully orchestrating whole cell activity to focus on a specific sought outcome can prove to be a particularly daunting task for genetic engineering because of the manifold interrelationships that characterize native patterns of metabolism that have evolved over extended periods of time [283]. A specific interest in this whole cell approach was triggered by the relatively poor performance of the PDH-CHMO in vitro system for ε-caprolactone formation [272,273]. Commencing in 2015, a progressive series of in vivo whole cell initiatives has been used to progressively improve the production of ε-caprolactone and its methyl-substituted derivative, including the inclusion of a lipase-catalysed terminal step to oligomerize the toxic lactone product [284,285,286,287,288]. In a related in vivo initiative, attention was also directed to the development of a recombinant strain of *E. coli* coexpressing an ADH, a BVMO, and an additional enoate reductase in an endeavour to extend the artificial metabolic cascade to include (*1S*,*1R*)-carveol as the test substrate (Figure 24 [289]).

The outcomes of this whole cell trial were encouraging, although, as might be expected from such an in vivo system, there was some evidence of product inhibition of the ADH, and/or back reaction by the ADH, and/or substrate inhibition of the enoate reductase. Encouraged by the outcome of this ‘systems biocatalysis’ approach, the pathway was then extended to include limonene as the feedstock [290], inspiring others to engineer an equivalent whole cell process that served to generate enantiocomplementary lactones [291]. Various other recombinant strains coexpressing different combinations of various cytochrome P450 monooxygenases, ADHs/enoate reductases, BVMOs, plus FDH to enhance NAD(H) cofactor recycling have also been trialled [292,293,294].

Around the same time that the Greifswald studies on redox-neutral closed-loop coenzyme recycling were progressing, research initiated in the Netherlands introduced a new twist to the in vitro artificial biocatalytic cascade concept. Capitalising on ADH-catalysed oxidative lactonisation methodology successfully developed decades earlier by J. Brian Jones [295], initial studies used HLADH to catalyse the substrate-coupled biotransformation of (*rac*)-2-phenyl-1-propanal and 1,4-butanediol, respectively, to a combination of the corresponding (*S*)-alcohol in high enantiomeric excess plus an initially formed C4-aldehyde (Figure 25 [296]). The aldehyde, being chemically unstable, then spontaneously cyclised to the equivalent hemiacetal, which itself was a viable substrate for biooxidation by HLADH to γ-butyrolactone, thus yielding a second stable recoverable product. This cascade of C4 interconversions resulted in 1,4-butanediol being dubbed a ‘smart cosubstrate’. However, unlike the previously developed in vitro Exeter-I and Exeter-II initiatives, this was not a closed-loop system replete with consequential downstream processing advantages. Acknowledging this significant strategic advantage of the established Exeter technology, a modification of the smart cosubstrate concept was then devised. It was reasoned that deploying cyclohexanone and 1,6-hexanediol as the cosubstrates with dually competent ADH and BVMO biocatalysts sharing complementary cofactor dependencies to promote coupled-enzyme cofactor recycling should, in principle, result in a redox-neutral convergent biocatalytic cascade that generated ε-caprolactone as the sole end product (Figure 26 [297]). While logistically more complex than the redox-neutral in vitro systems previously developed at Exeter, this convergent biocatalytic cascade did share the advantage of not generating a coproduct, thereby simplifying downstream processing. Testing various ADHs and CHMOs then led to the adoption of a system comprising the NADPH-dependent CHMO from NCIB 9871, and the NADP(H)-dependent alcohol de-hydrogenase from *Thermoanaerobium ethanolicus* (TeSADH). Scaling up (50 mL) of the redox-neutral coupled-system resulted in a confirmed >99% yield of ε-caprolactone within 18 h. As a consequence of this further success, 1,6-hexanediol was dubbed with the accolade of a ‘double-smart cosubstrate’. However, dependence of the system on CHMO from NCIB 9871 was viewed as problematic, both because of the limited stability of the wild-type enzyme and its susceptibility to both substrate and product inhibition [230]. The stability issue was addressed firstly by testing the performance of AFL706, a newly described fungal CHMO [298], and secondly subjecting the wild-type CHMO from NCIB 9871 to site-directed mutagenesis to generate M15 DS and M16 DS, which both incorporated a reported intramolecular disulphide bridge [299]. While AFL706 proved to be more resistant to inhibition than wild-type CHMO, it had disappointing low specific activity. However, when tested as purified proteins, both M15 DS and M16 DS proved more resistant to inhibition than wild-type CHMO, leading to a significantly increased yield of ε-caprolactone. The potential of solvent engineering to mitigate inhibition of wild-type CHMO from NCIB 9871 was also investigated. A study designed to minimise inhibition resulting from the accumulating lactone by including dodecane in an equivalent two-liquid phase system (2LPS) was tested, but with only minimal improvement in performance [300]. However, a further trial undertaken with an aqueous–isooctane biphasic system and with TeSADH exchanged for the alcohol dehydrogenase from *Lactobacillus kefir* DSM 202587 was found to generate ε-caprolactone at an enhanced rate [301].

The development by the Netherlands team of an equivalent in vitro NAD(H)-dependent redox-neutral convergent cascade system was seen an important next step to expand its applicability [302]. One potential solution involved replacing wild-type NADPH-dependent CHMO with an NADH-compatible 3M variant of CHMO generated by site-directed mutagenesis [303]; however, the relatively poor performance of the variant ruled this option out. Attention then turned to deploying other suitable NADH-dependent monooxygenases to catalyse the cyclohexanone branch of the convergent cascade. While a type II flavin monooxygenase (FMO) originally isolated from the marine isolate *Stenotrophomonas maltophilia* PML 168 [304,305] was reported to biotransform (*rac*)-fused ring cyclobutanones to corresponding lactones by both NADH- and NADPH-dependent biooxygenations, the activity of the enzyme with NADH was relatively low [306,307]. Consequently, other type II FMOs more active with NADH were sought in preference. Screening focused on FMO-E, FMO-F, and FMO-G, three type II FMOs isolated from *Rhodococcus jostii* RHA 1 [308,309]. After testing with a range of mono- and bicyclic ketones, including (*rac*)-bicyclo[3.2.0]hept-2-en-6-one, FMO-E along with NADH-dependent HLADH were chosen to develop two redox-neutral NADH-dependent convergent biocatalytic cascades [302]. One system comprised the previously used cosubstrate 1,4-butanediol in a new combination with cyclobutanone, that was developed to generate 130 mM γ-butyrolactone from 100 mM cyclobutanone and 50 mM 1,4-butanediol. The other more radical system comprised (*rac*)-bicyclo[4.2.0]octan-7-one and cis-1,2-cyclohexanedimethanol as cosubstrates, and was successfully developed to yield a 2.3:1 mixture of the corresponding chiral regioisomeric (3a*S*,7a*S*)-2-oxa-and (3a*R*,7a*S*)-3-oxa-lactones (e.e. 74–89% and >99% respectively, Figure 27). By comparing the data generated from the different tested in vitro NAD(P)H-dependent ADH + BVMO convergent cascades, it was apparent that the best overall performance was delivered by the NADP(H)-dependent system comprising the MS 16 DS double mutant of CHMO from NCIB 9871 plus TeSADH. Progress curve analysis [310] was then used to develop iteratively the outcomes from two kinetic model systems differing in the titres of both M16 DS (Model 1 = 0.033 mM: Model 2 = 0.025 mM) and TeSADH (Model 1 = 0.00026 mM: Model 2 = 0.00013 mM). Model 2 generated the better predicted outcomes, with root-mean-square-error (RMSE) values approximately half those predicted by Model 1.

It is fitting that a review tracing the 40-year history of redox biotransformations of bicyclo[3.2.0]carbocyclic molecules should conclude with a focus on recent research detailing the BVMO-catalysed generation of a chiral bicyclo[3.3.0]lactone as the key initiating reaction for the subsequent chemoenzymatic synthesis of both natural and analogue prostaglandins [1]. The published paper is founded on historic concepts of prostaglandin synthesis that can be related back directly to research undertaken by Glaxo Group Research in the 1980s [107], and even further back to the contributions either directly made by [11] or inspired by [12], E.J.Corey (vide supra). The multi-centre Chinese team planned a retrosynthetic route dependent on the recognised potential of halo-substituted-bicyclo[3.2.0]alkenones as valuable synthons because of the potential different opportunities offered by each ring for subsequent stereocontrolled modifications [13,14]. The key initial step introduced by Zhu et al. was the BVMO-catalysed regio- and stereospecific ring expansion of (*rac*)-7,7-dichlorobicyclo[3.2.0]hept-2-en-6-one to the corresponding chiral γ-lactone (*3aR*,*6aS*)-3,3-dichloro-3,3a,6,6a-tetrahydro-2H-cyclopenta[*b*]furan-2-one (Figure 28). Four previously reported BVMOs (CHMO_Rhodo_ [311], CHMO_Arthro_ [311], CHMO_Brevi-1_ [312], and CHMO-MO14 [313]), all sourced from microbial environmental isolates, were screened for the ability to lactonise the relevant (*rac*)-7,7-disubstituted-bicyclo[3.2.0]ketone. The best-performing candidate proved to be CHMO_Rhodo_ (Table 19).

Capitalising directly on Walsh and Whiteside’s 1989 precedent [161], an in vitro coupled-enzyme NADP(H)-recycling system incorporating CHMO_Rhodo_ with glucose-6-phosphate plus glucose as the sacrificial substrate proved valuable in producing larger amounts of the sought 2-oxa-lactone with absolute regio- and stereospecificity (e.e. 99%). The lactone synthon was then processed further via a progressive sequence of chemical reactions that served to generate four related 3-enone-substituted lactones. An in vitro bioreduction step was then used to promote diastereoselective reduction to the corresponding allylic alcohol-substituted lactones (Figure 29). The biocatalyst used was a ketoreductase (KRED) isolated from the yeast *Pichia anomala* [314], which was deployed along with glucose dehydrogenase and glucose as an NADP(H) cofactor recycling system. The recovered allylic alcohols were then in turn chemically converted into the final end products cloprostenol, fluprostenol, bimatoprost, travoprost, and prostaglandin F_2α_. (Figure 29). Zhu et al.’s 2021 deployment of the biooxygenation of a bicyclo[3.2.0]carbocyclic ring system as the key initiating step in the chemoenzymatic synthesis of PGF_2α_ provides a striking contrast to the exclusively chemocatalytic synthesis of the same molecule from cyclopentadiene reported by E.J. Corey over five decades previously in 1969, but reflects well on the proven value of redox-neutral BVMO-dependent biocatalysis in the 21st century.

## Figures and Tables

**Figure 1 molecules-28-07249-f001:**
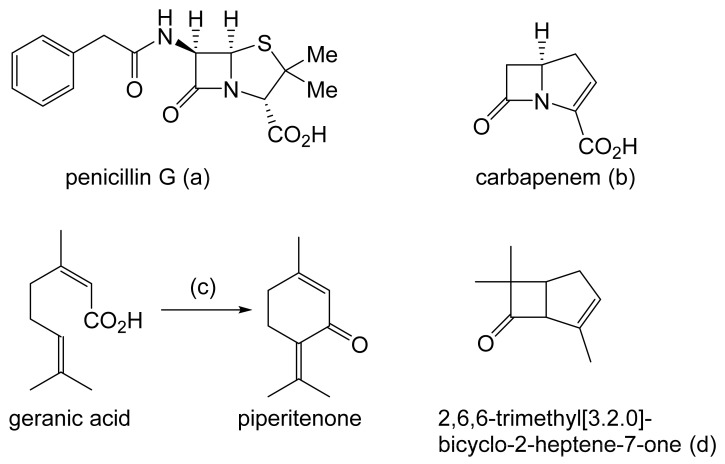
(**a**–**d**) Early (pre-1963) examples of known molecules with the characteristic bicyclo[3.2.0] ring structure.

**Figure 2 molecules-28-07249-f002:**
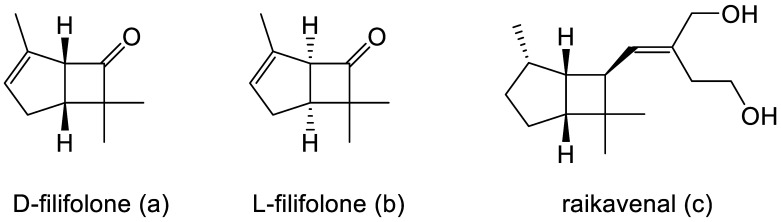
(**a**–**c**) Characterised natural metabolites with the characteristic bicyclo[3.2.0]carbocyclic ring structure.

**Figure 3 molecules-28-07249-f003:**
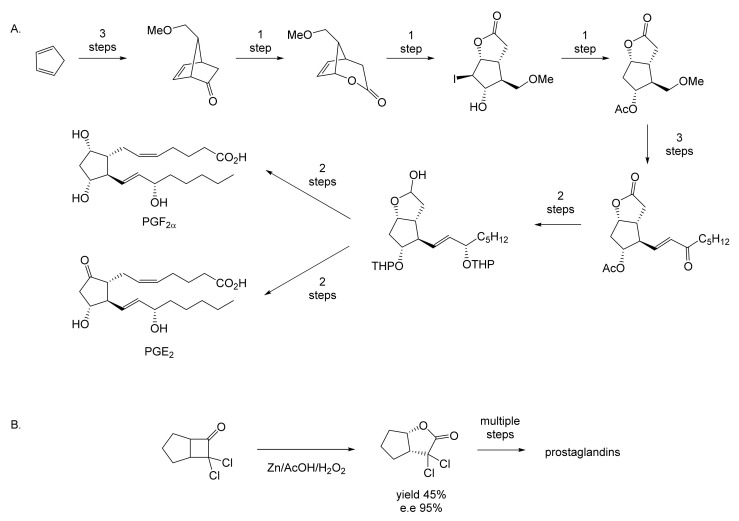
Key steps in the chemocatalytic synthesis of prostaglandins: (**A**) Corey et al. [11]; (**B**) Tomoskozi et al. [12].

**Figure 4 molecules-28-07249-f004:**
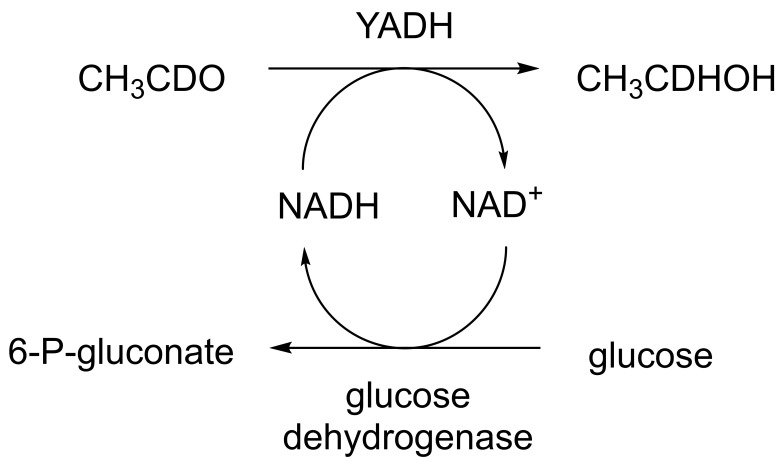
First reported example of coupled-enzyme cofactor recycling [31].

**Figure 5 molecules-28-07249-f005:**
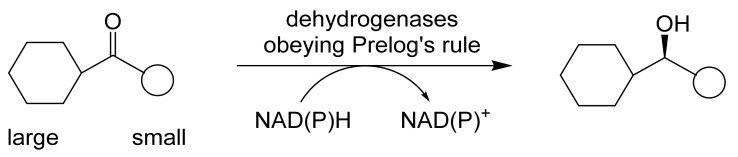
Prelog’s predictive rule for bioreduction by alcohol dehydrogenases. Adapted from [59].

**Figure 6 molecules-28-07249-f006:**
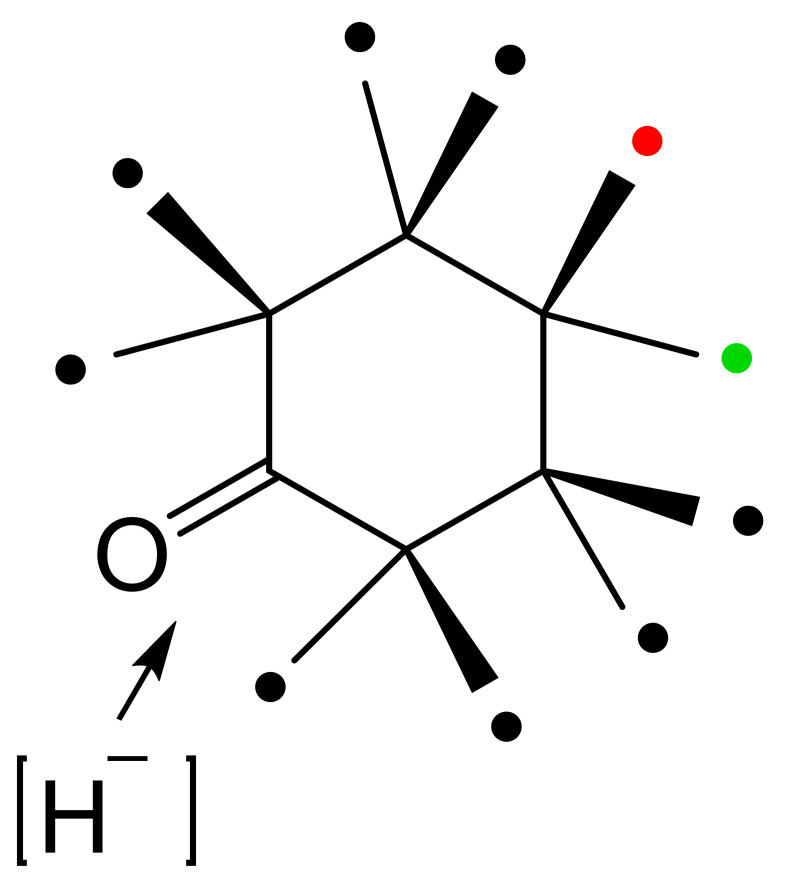
Predictive outcome model developed for HLADH and substituted cyclohexanone substrates. The position of substituents relative to the carbonyl oxygen atom was found to be either highly favourable (🔴), favourable, (🟢), or disfavourable (⚫). Adapted with permission from [78]. 1982. Royal Society of Chemistry.

**Figure 7 molecules-28-07249-f007:**
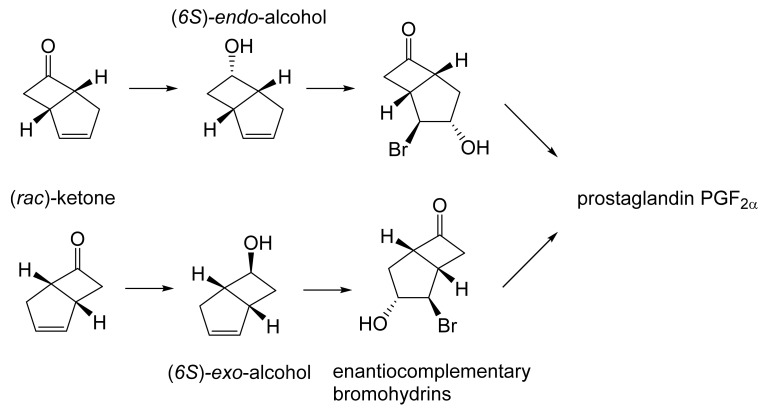
Bioreduction of (*rac*)-bicyclo[3.2.0]hept-2-en-6-one by Fermipan Bakers’ yeast. Adapted with permission from [81]. 1979. Royal Society of Chemistry.

**Figure 8 molecules-28-07249-f008:**
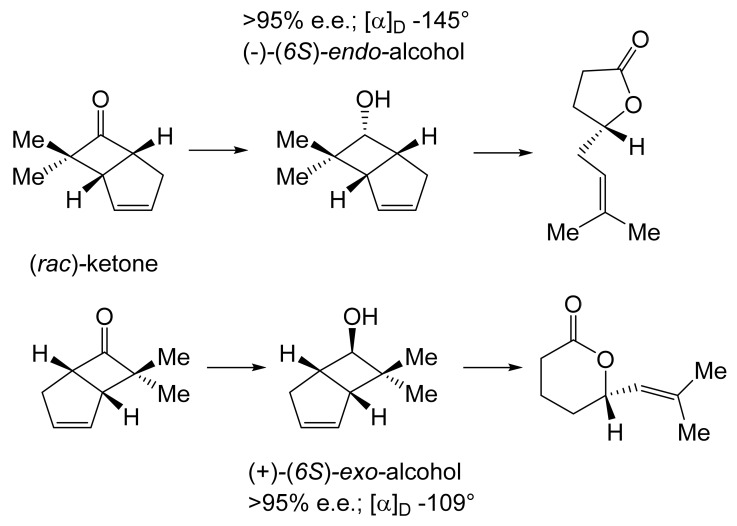
Bioreduction of (*rac*)-7,7-dimethylbicyclo[3.2.0]hept-2-en-6-one by *Mortierella ramanniana.* Adapted with permission from [103]. 1984. Royal Society of Chemistry.

**Figure 9 molecules-28-07249-f009:**
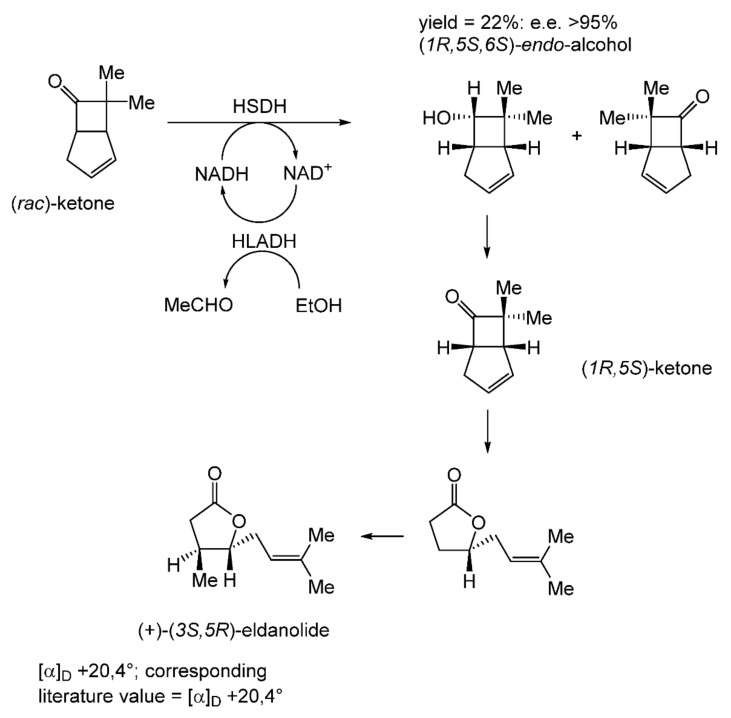
Coupled-enzyme biotransformation of (*rac*)-7,7-dimethylbicyclo[3.2.0]hept-2-en-6-one to the chiral (*1R*,*5S*)-ketone, which serves as a synthon for (+)-(*3S*,*5R*)-eldanolide. Adapted with permission from [112]. 1987. Royal Society of Chemistry.

**Figure 10 molecules-28-07249-f010:**
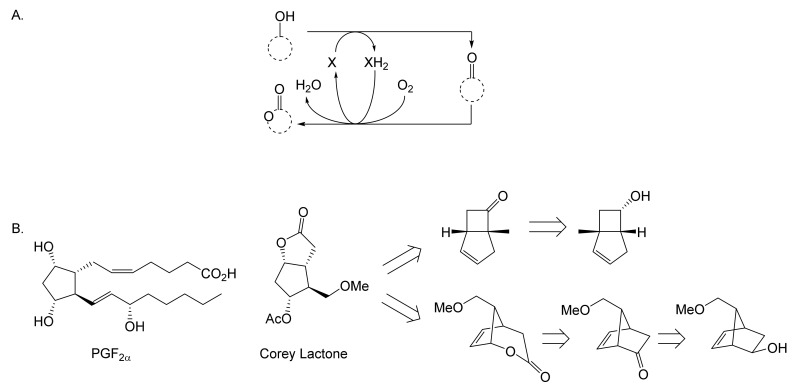
(**A**) outline of the Exeter BVMO-based lactone strategy (an ADH + BVMO linear biocatalytic cascade with closed-loop cofactor recycling); (**B**) envisaged potential applications (including Corey lactones and prostaglandins) of the Exeter BVMO-based lactone strategy.

**Figure 11 molecules-28-07249-f011:**
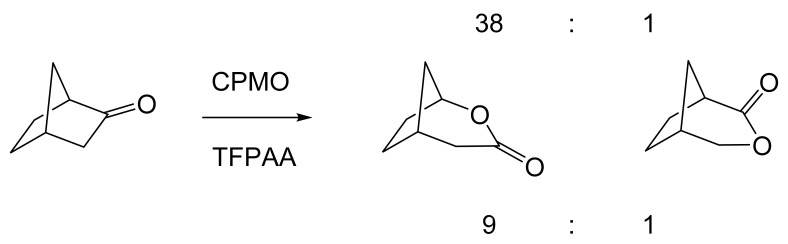
Outcome of the biooxygenation of norbornanone by a washed cell suspension of cyclopentanol-grown *Pseudomonas* NCIB 9872 compared to the equivalent chemical oxidation by trifluoroperacetic acid. Adapted with permission from [187]. 1989. Springer Nature.

**Figure 12 molecules-28-07249-f012:**
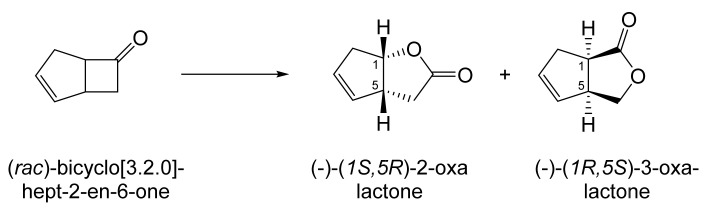
Biooxygenation of (*rac*)-bicyclo[3.2.0]hept-2-en-6-one by cyclohexane-1,2-diol-grown *Acinetobacter* TD63. Reprinted with permission from [188]. 1989. Elsevier.

**Figure 13 molecules-28-07249-f013:**
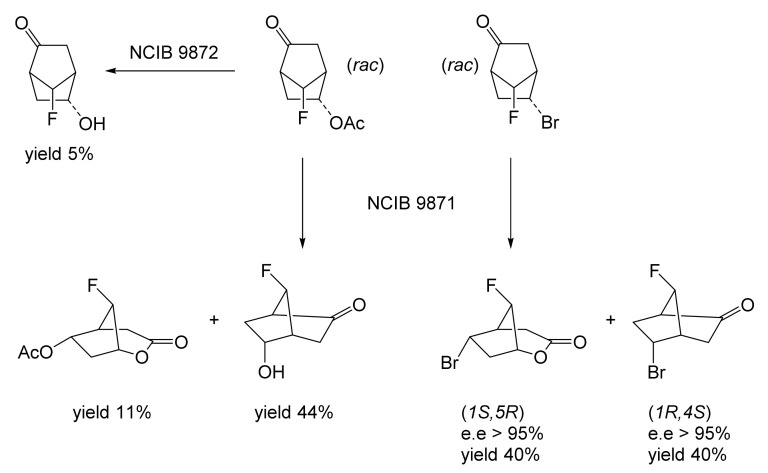
Biooxygenation of (*rac*)-5-acetoxy-7-fluorobicyclo[2.2.1]heptan-2-one by washed cell suspensions of cyclohexanol-grown *Acinetobacter* NCIB 9871 and cyclopentanol-grown *Pseudomonas* NCIB 9872 [189].

**Figure 14 molecules-28-07249-f014:**
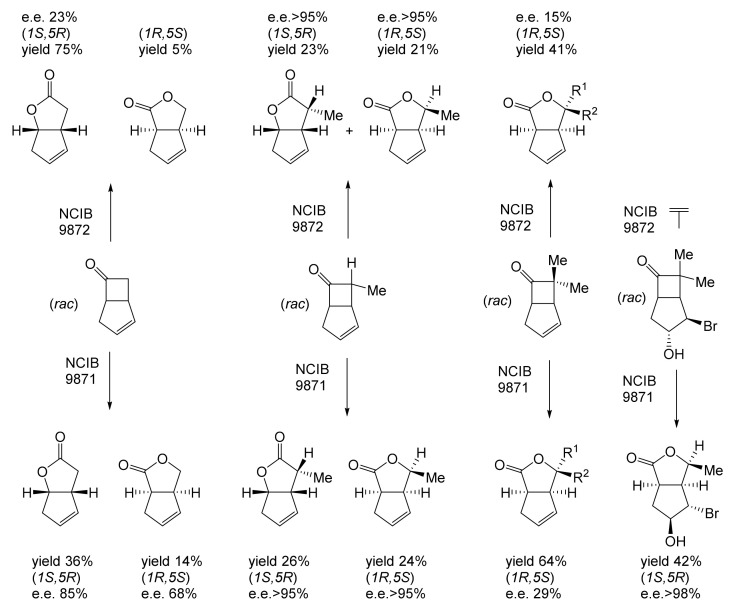
Comparison of the outcomes of biooxygenations of unsubstituted and various substituted (*rac*)-bicyclo[3.2.0]hept-2-en-6-ones by washed cell suspensions of cyclohexanol-grown *Acinetobacter* NCIB 9871 and cyclopentanol-grown *Pseudomonas* NCIB 9872. Adapted with permission from [191]. 1990. Royal Society of Chemistry.

**Figure 15 molecules-28-07249-f015:**
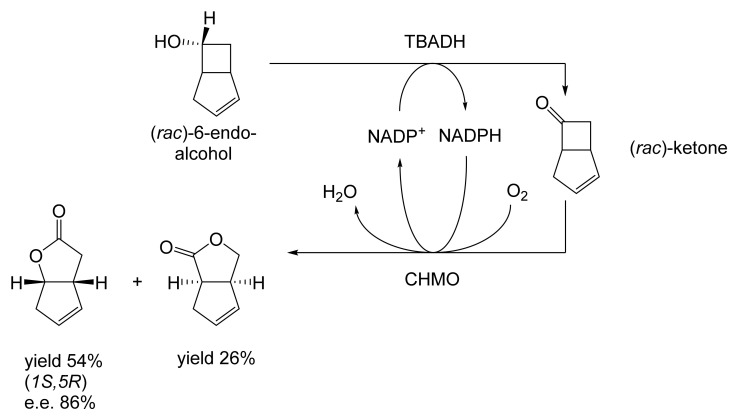
Biooxygenation of (*rac*)-6-*endo*-bicyclo[3.2.0]hept-2-en-6-ol by *Thermoanaerobium brockii* ADH plus cyclohexanone monooxygenase, the original ‘Exeter-I system’ [202]. Reprinted with permission from [2]. 1991. Royal Society of Chemistry.

**Figure 16 molecules-28-07249-f016:**
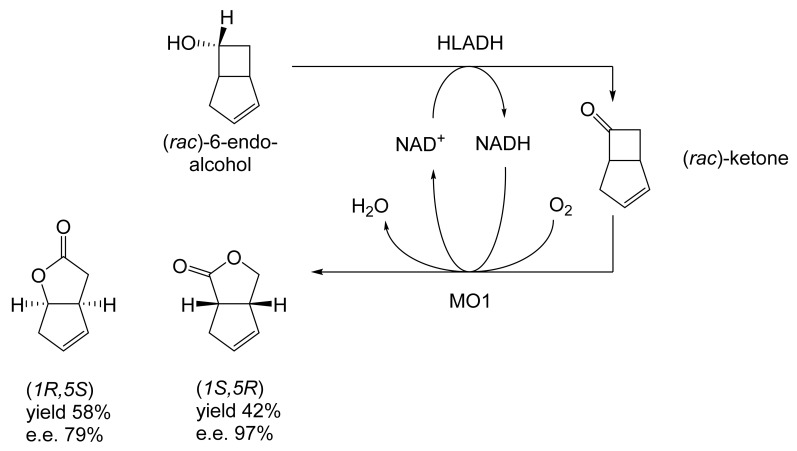
Biotransformation of (*rac*)-6-endo-bicyclo[3.2.0]hept-2-en-6-ol by horse liver ADH plus cyclohexane monooxygenase, the original Exeter-II system [202]. Reprinted with permission from [212]. 1994. Royal Society of Chemistry.

**Figure 17 molecules-28-07249-f017:**
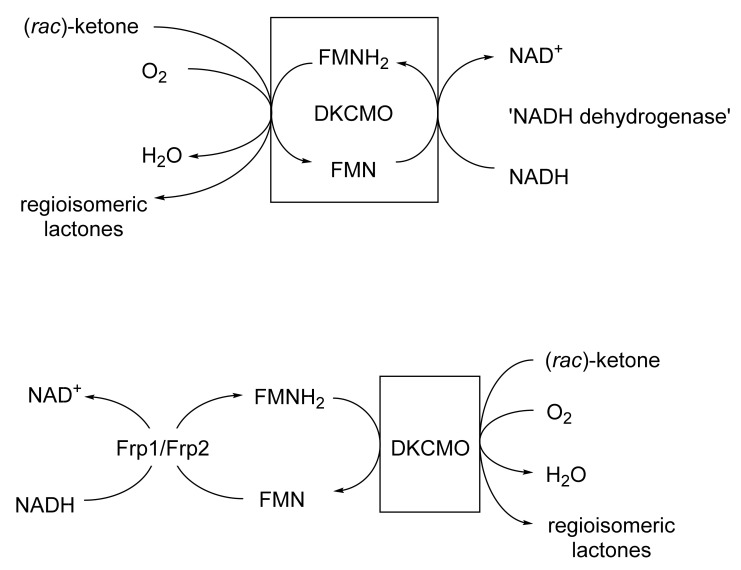
Comparison of the historical mode of action of the DKCMOs proposed by Gunsalus et al. (‘NADH dehydrogenase-dependent’ [170]), and the currently accepted Frp1/Frp2-dependent revised mode of action as fd-TCMOs [173,177].

**Figure 18 molecules-28-07249-f018:**
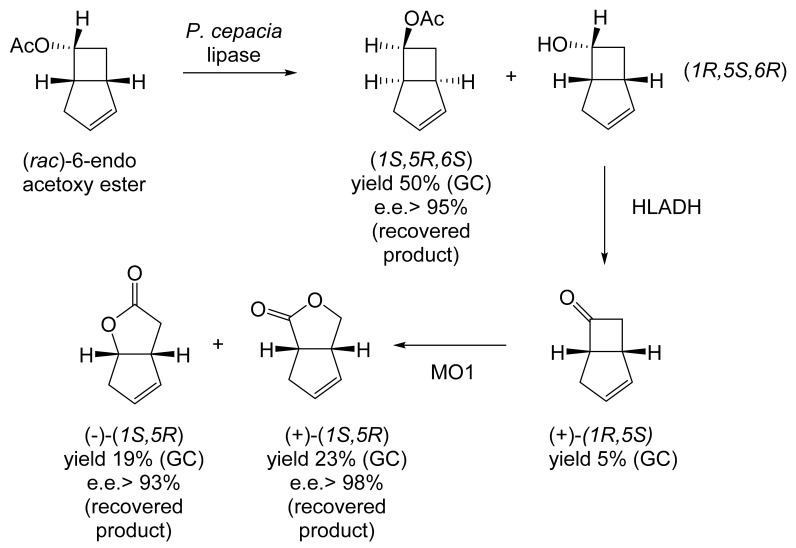
*P. cepacia* lipase + horse liver ADH + the isoenzymic DKCMOs (MO1) operating as a three-enzyme artificial linear cascade to biotransform (*rac*)-6-*endo*-acetoxybicyclo[3.2.0]hept-2-ene. Adapted with permission from [212]. 1994. Royal Society of Chemistry.

**Figure 19 molecules-28-07249-f019:**
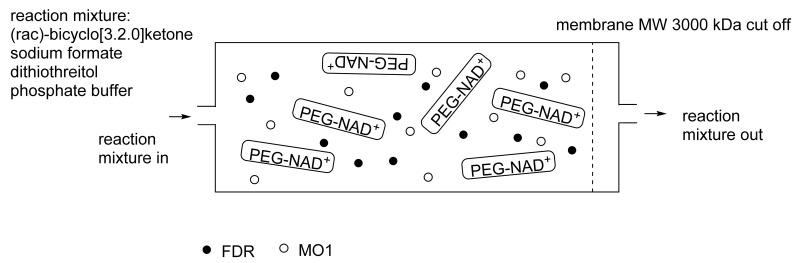
Membrane bioreactor (PEG-NAD^+^ + formate dehydrogenase + MO1) used to biooxygenate (*rac*)-bicyclo[3.2.0]hept-2-en-6-one for 12 successive 24 h-cycles [218].

**Figure 20 molecules-28-07249-f020:**
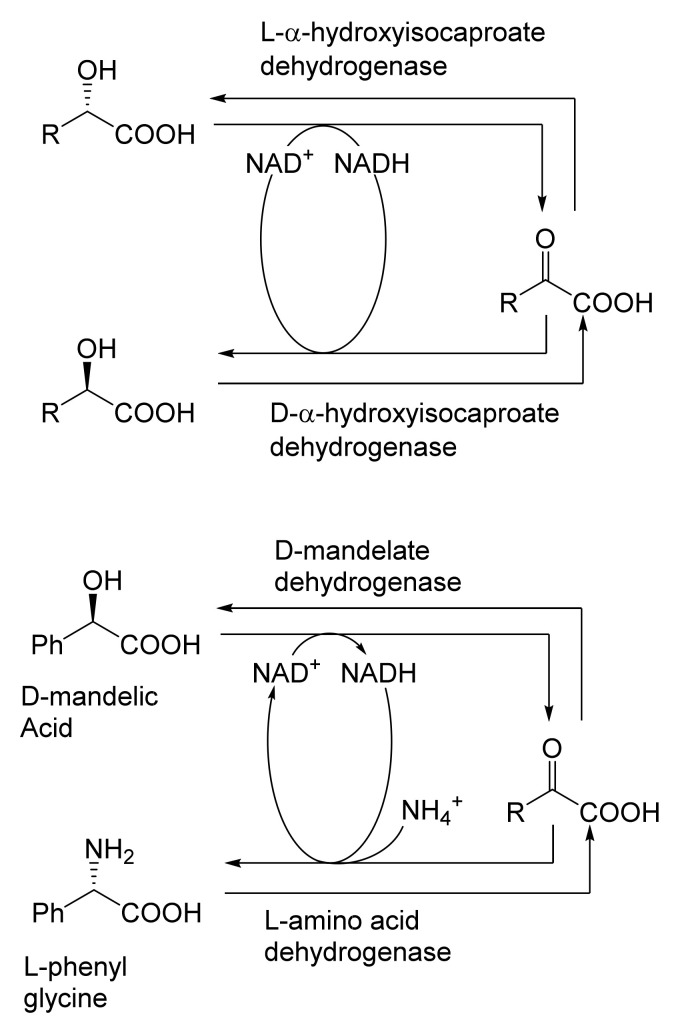
Redox-neutral closed-loop cofactor recycling by pairs of stereocomplementary alcohol dehydrogenases. Adapted with permission from [264,265]. 2009, Elsevier, and 2010, John Wiley and Sons, respectively.

**Figure 21 molecules-28-07249-f021:**
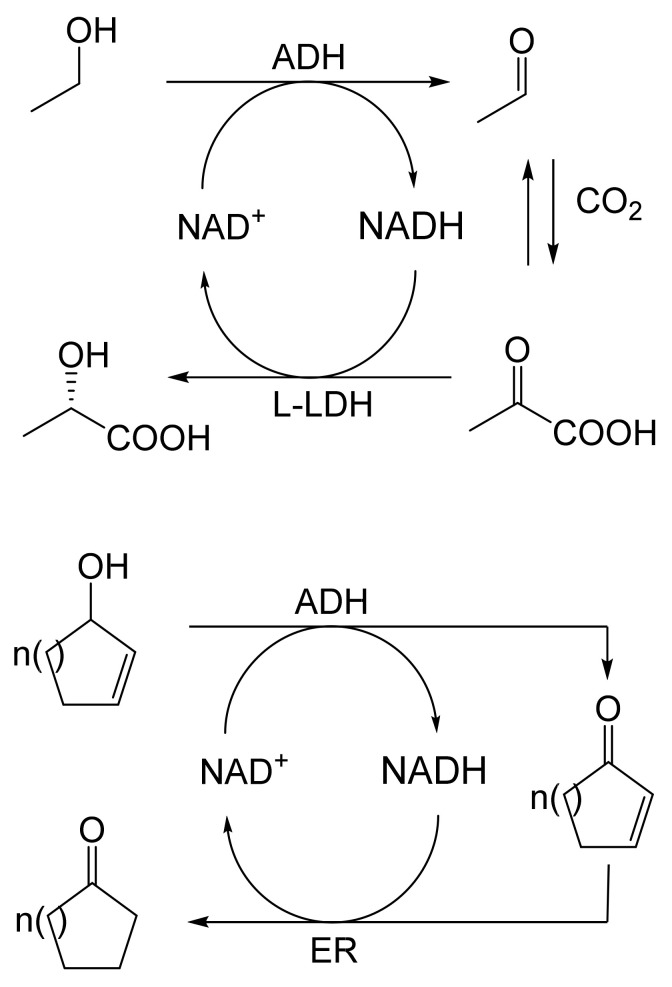
Redox-neutral closed-loop cofactor recycling by pairs of complementary enzymes. Adapted with permission from [266,267]. 2010, John Wiley and Sons, and 2012, Royal Society of Chemistry, respectively.

**Figure 22 molecules-28-07249-f022:**
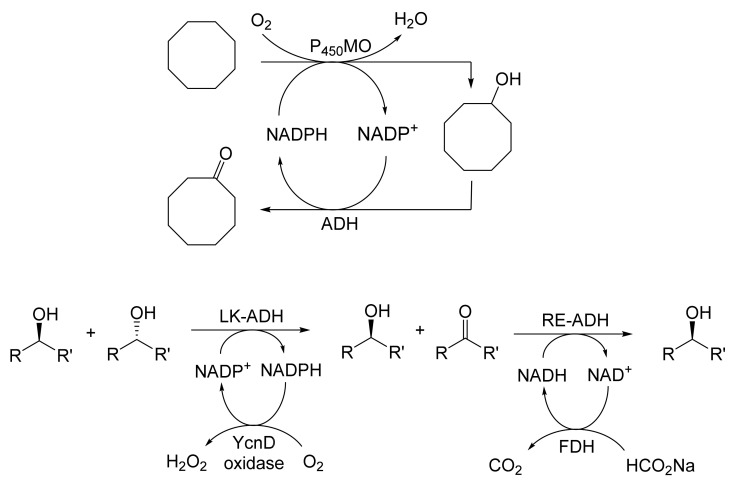
A simple redox-neutral closed-loop cascade developed with CPY BM3 and an alcohol dehydrogenase. Adapted with permission from [269]. 2013. John Wiley and Sons. A more complicated conventional system dependent on two discrete consecutive coupled-enzyme systems for the recacemisation of secondary alcohols. Adapted with permission from [274]. 2008. American Chemical Society.

**Figure 23 molecules-28-07249-f023:**
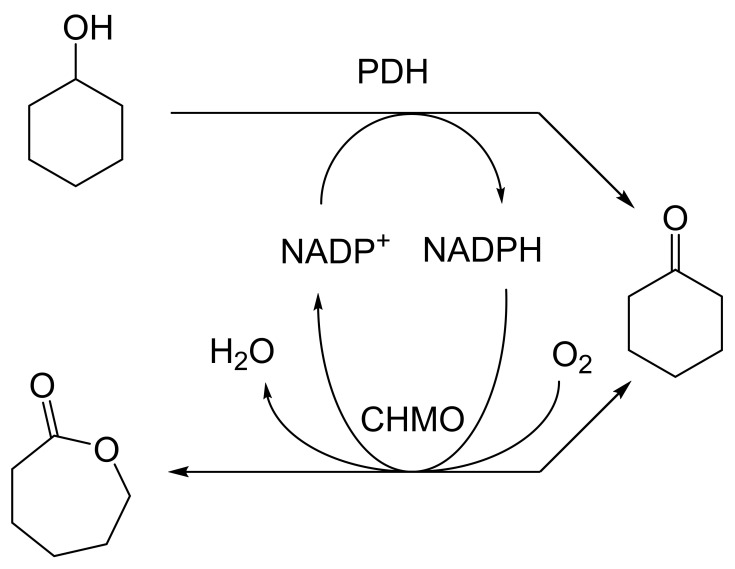
Redox-neutral in vitro linear biocatalytic cascade with closed-loop cofactor recycling for ε-caprolactone synthesis. Reprinted with permission from [272]. 2013. Elsevier.

**Figure 24 molecules-28-07249-f024:**
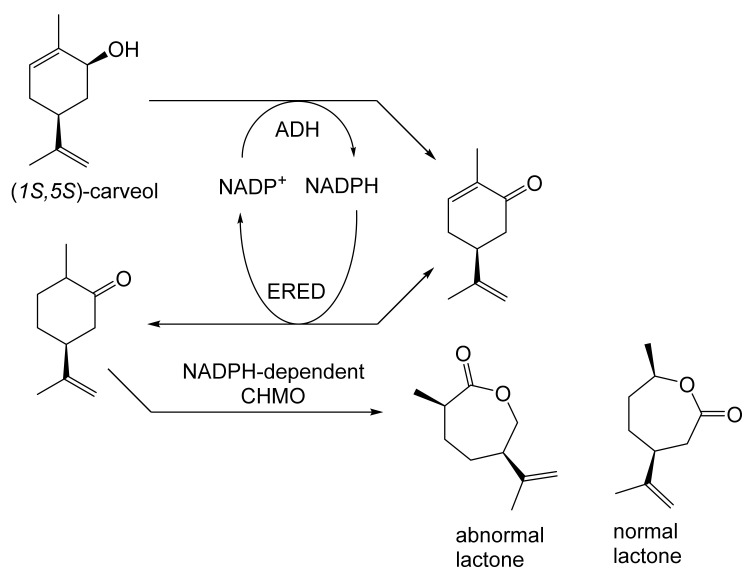
Biotransformation of (*1S*,*5S*)-carveol deploying a three-enzyme in vivo biocatalytic linear cascade with closed-loop cofactor recycling. Adapted with permission from [289]. 2013. John Wiley and Sons.

**Figure 25 molecules-28-07249-f025:**
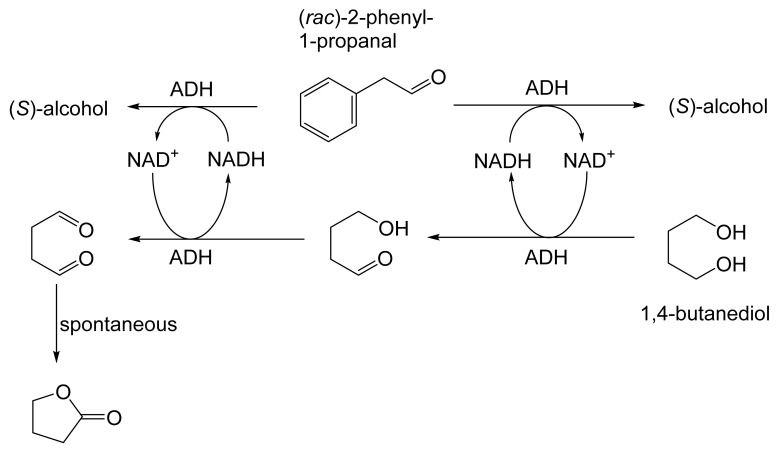
The ‘smart cosubstrate’ artificial biocatalytic cascade concept. Adapted from [296].

**Figure 26 molecules-28-07249-f026:**
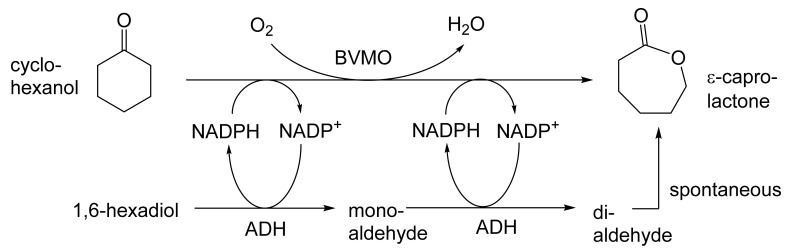
The ‘double-smart cosubstrate’ convergent biocatalytic cascade concept. Adapted with permission from [297]. 2015. John Wiley and Sons.

**Figure 27 molecules-28-07249-f027:**
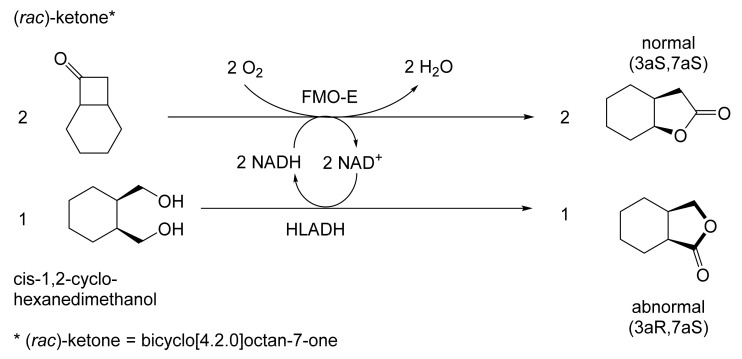
The ‘double-smart cosubstrate’ convergent biocatalytic cascade concept for the synthesis of chiral lactones from (*rac*)-bicyclo[4.2.0]octan-7-one and cis-1,2-cyclohexanedimethanol. Reprinted with permission from [302]. 2017. John Wiley and Sons.

**Figure 28 molecules-28-07249-f028:**
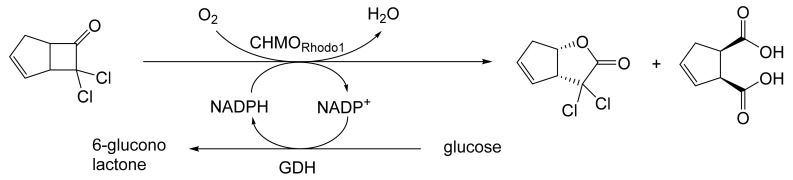
Key initial BVMO-catalysed redox-neutral coupled-enzyme cofactor recycling step involved in the chemoenzymatic synthesis of several prostanoids. Adapted from [1].

**Figure 29 molecules-28-07249-f029:**
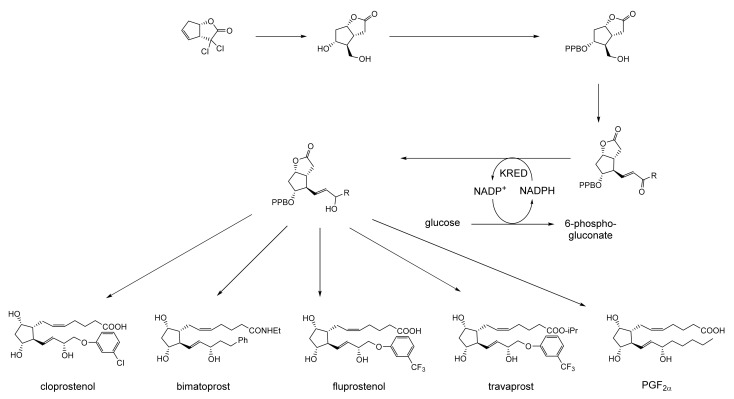
Coupled-enzyme bioreduction as a key intermediary step in the chemoenzymatic synthesis of cloprostenol, bimatoprost, fluprostenol, travaprost and PGF_2α_. Adapted from [1].

**Table 1 molecules-28-07249-t001:** Known substrate range of alcohol dehydrogenases (ADH/DH). YADH = yeast ADH; HLADH = horse liver ADH; TBADH = *Thermoanaerobium brockii* ADH; HSDH = hydroxysteroid DH; CPDH = cyclopentanol DH; CHDH = cyclohexanol DH.

ADHs	Aliphatic Aldhydes	Aliphatic Ketones	Monocyclic Ketones	Bicyclo-[2.2.1]- Ketones	Bicyclo-[3.2.0]- Ketones	Polycyclic Ketones
YADH	+	++	-	-	-	-
HLADH	-	++	++	++	++	+
TBADH	+	++	+++	+	+	-
HSDH	-	-	+	++	++	+++
CPDH	-	+	+++	+++	++	+
CHDH	-	+	+++	++	+++	+

**Table 2 molecules-28-07249-t002:** Screen by Glaxo Group Research of eukarytic microbial strains for the bioreduction of (*rac*)-bicyclo[3.2.0]hept-2-en-6-one. Adapted with permission from [89]. 1983. Royal Society of Chemistry.

Tested Strain	Glaxo Number	% Conversion to 6-alcohol (24 h)		Ratio
		*endo*-	*exo*-	*endo*-:*exo*-
Fermipan		7	3	2.3:1
*S.cerevisiae*	C50	4	2	2:1
*S.cerevisiae*	C1739	3	1	3:1
*Curvularia lunata*	C2100	22	trace	>22:1
*Mortierella ramanniana*	C2506	32	0	>32:1
*Rhodotorula rubra*	C1768	26	10	2.6:1

**Table 3 molecules-28-07249-t003:** Comparison of the bioreduction of (*rac*)-bicyclo[3.2.0]hept-2-en-6-one by growing cultures of *Curvularia lunata* C2100 and *Mortierella ramanianna* C2056 [89].

Sample Time (min) Post-Inoculation	% Conversion (*rac*)-ketone	
	*C. lunata*	*M. ramanianna*
10	0.5	2
20	1	3
40	2	6
80	6	9
150	12	14
300	12	18

**Table 4 molecules-28-07249-t004:** Comparison of the bioreduction of (*rac*)-bicyclo[3.2.0]hept-2-en-6-one (test ketone 1) and (*rac*)-7,7-dimethylbicyclo[3.2.0]hept-2-en-6-one (test ketone 2) by commercial preparations of horse liver alcohol dehydrogenase (HLADH), and the alcohol dehydrogenases from *Thermoanaerobium brockii* (TBADH) and *Streptomyces hydrogenans* (HSDH). Adapted with permission from [60]. 1985. Elsevier.

Test Ketone	ADH	Cofactor Recycling Method		6-*endo*-alcohol Product Formed	
		(Coupled- Substrate)	(Coupled- Enzyme)	+/−	e.e.%
(*rac*)-bicyclo[3.2.0]-	HLADH	NADH (ethanol)		+	<10%
hept-2-en-6-one					
	HSDH		NADH (G-6-DH)	+	<10%

	TBADH	NADPH (propan-2-ol)		+	>95%

(*rac*)-7,7-dimethyl- bicyclo[3.2.0]hept- 2-en-6-one	HLADH	NADH (ethanol)		none detected	
	TBADH	NADPH (propan-2-ol)		none detected	
	HSDH		NADH (HLADH)	+	>95%

			NADH (G-6-DH)	+	>95%


**Table 5 molecules-28-07249-t005:** Bioreduction in aqueous ethanol of various substituted (*rac*)-bicyclo[3.2.0]hept-2-en-6-ones by hydroxysteroid dehydrogenase (coupled-substrate system), and in combination with yeast alcohol dehydrogenase (coupled-enzyme system). 1 = 7,7-dihydroheptenone; 2 = 7-*endo*-chloroheptenone; 3 = 7,7-dichloroheptenone; 4 = 7,7-dimethylheptenone; 5 = 7.7-*exo*-chloro-*endo*-methylheptenone; 6 = 7,7-*endo*-chloro-*exo*-methylheptenone [61].

Tested Substituted (*rac*)-Bicyclo[3.2.0]hept-2-en-6-one Ketone						
	1	2	3	4	5	6
Single-phase couple-substrate						
*K_m_*	80	7	1.8	50	10	4
*V_max_*	18	86	71	36	36	71
[α]^D^	+4.7	−190	−155	−145	−377	−151
Single-phase coupled-enzyme						
e.e. %	8	>90	>95	>95	n.d.	n.d.
chirality	n.d.	(*6S*)-	(*6S*)-	(*6S*)-	n.d.	n.d.

**Table 6 molecules-28-07249-t006:** Bioreduction of substituted (*rac*)-7,7-dichloro[3.2.0]hept-2-en-6-ones by a hydroxysteroid dehydrogenase plus yeast alcohol dehydrogenase two-phase coupled-enzyme system tested using a range of different representative solvents as the organic phase [61].

Organic-Phase Solvent	Organic Solvent Parameters		% Bioreduction of the (*rac*)-Ketone by HSDH-YADH Coupled- Enzyme System in Two-Phase Media
	Polarity	Dielectric	
	Index	Constant	
None			68
Octanol	2.4	10.3	34
Hexane	0.1	1.9	25
Chloroform	4.1	4.8	12
Ethyl acetate	4.4	6.0	12
Dichloromethane	3.5	10.0	1

**Table 7 molecules-28-07249-t007:** Biooxidation of (*rac*)-6-*endo*-bicyclo[3.2.0]hept-2-en-6-ol by recovered cells of *Bacillus stearothermophilus.* Adapted with permission from [118]. 1994. Elsevier.

Whole Cell Biocatalyst	Single-Phase System	Time (h)	Ketone		
			Yield %	e.e.%	Chirality
Washed cells	Water	6	41	99	(*1S*,*5R*)
Washed cells	Heptane	6	49	>99	(*1S*,*5R*)
Immobilised cells	Heptane	3	41	>99	(*1S*,*5R*)

**Table 8 molecules-28-07249-t008:** Bioreduction of substituted (*rac*)-bicyclo[3.2.0]heptenones by a washed cell suspension of *Saccharomyces cerevisiae* compared to the biooxidation of equivalent heptenols by a washed cell suspension of *Bacillus stearothermophilus*. Adapted with permission from [124]. 1994. Elsevier.

Test Substrate	Biocatalyst (Washed Cells)	Recovered Products		
		Yield %	e.e.%	Chirality
(*rac*)-4-methyl- bicyclo[3.2.0]- heptenone	Bakers’s yeast	30 17	96 >99	(*6S*)-*endo*-ol (*6S*)-*exo*-ol
(*rac*)-1,4-dimethyl bicyclo[3.2.0]- heptenone	Bakers’s yeast	37 47	>99 >99	(*6S*)-*endo*-ol (*6S*)-*exo*-ol
(*rac*)-4-methyl- bicyclo[3.2.0]- heptenol	*Bacillus* *stearothermo-* *philus*	45 45	82 95	(*1R*,*5S*)-one (*6R*)-*endo*-ol
(*rac*)-1,4-dimethyl bicyclo[3.2.0]- heptenol	*Bacillus* *stearothermo-* *philus*	47 47	86 98	(*1R*,*5S*)-one (*6R*)-*endo*-ol

**Table 9 molecules-28-07249-t009:** Biotransformation of (*rac*)-bicyclo[3.2.0]hept-2-en-6-ol and (*rac*)-bicyclo[3.2.0]hept-2-en-6-one by a crude cell-free extract of *Bacillus steaothermophilus* in water and a 9:1 water:heptane mixture. Adapted with permission from [126]. 1996. Elsevier.

Bicyclic Test Substrate	Time (h)	Medium	Products		
			Yield %	e.e.%	Chirality
(*rac*)-*endo*- bicyclo[3.2.0]- hept-2-en-6-ol	18	water	43	98	(*1S*,*5R*)-one
			52	92	(*6R*)-*endo*-ol
	24	water	52	90	(*1S*,*5R*)-one
			47	97	(*6R*)-*endo*-ol
	48	water- heptane	28 47	>99 97	(*1S*,*5R*)-one (*6R*)-*endo*-ol
(*rac*)-bicyclo- [3.2.0]-hept-2- en-6-one	36	water	54	62	(*1R*,*5S*)-one
			43	92	(*6S*)-*endo*-ol
	72	water- heptane	72 24	26 >99	(*1R*,*5S*)-one (*6S*)-*endo*-ol

**Table 10 molecules-28-07249-t010:** Bioreduction by growing cultures of various yeasts and fungi of (*rac*)-bicyclo[3.2.0]hept-2-en-6-one (ketone 1), (*rac*)-4-methylbicyclo[3.2.0]hept-2-en-6-one (ketone 2), (*rac*)-1-methylbicyclo[3.2.0]hept-2-en-6-one (ketone 3), and (*rac*)-1,4-dimethylbicyclo[3.2.0]hept-2-en-6-one (ketone 4). Adapted with permission from [127]. 1996. Elsevier.

Bicyclic Test Ketone	Whole Cell Biocatalyst	Recovered Products			
		(*6S*)-*endo*	(*6R*)-*endo*	(*6R*)-*exo*	(*6S*)-*exo*
		Yield% e.e.%	Yield% e.e.%	Yield% e.e.%	Yield% e.e.%
	*S. cerevisiae* RM1		27 70		4 38
	*S. cerevisiae* RM9	20 60			16 94
	*S. cerevisiae* ML31		52 16		3 >99
(*rac*)-1	*Kluy. lactis*	32 76			5 8
	*Pen. digitalum*	21 62			8 >99
	*Rhiz. nigricans*	22 66			7 >99
	*Trichoderma* sp.	32 72			3 >99
	*S. cerevisiae* RM1	60 60			
(*rac*)-2	*S. cerevisiae* RM9	54 50			9 >99
	*Muc. spirescens*	43 20			2 >99
	*Trichoderma* sp.	32 72			3 >99
	*S. cerevisiae* RM1	72 80			13 >99
(*rac*)-3	*S. cerevisiae* RM9	58 95			13 97
	*Muc. spirescens*	70 70			15 >99
	*Trichoderma* sp.	77 36			8 >99
	*S. cerevisiae* RM1	46 >99			12 >99
(*rac*)-4	*S. cerevisiae* RM74	3 >99			31 >99
	*Fusarium* sp.		28 38		7 >99
	*Trichoderma* sp.	16 98			21 >99

**Table 11 molecules-28-07249-t011:** Bioreduction of (*rac*)-bicyclo[3.2.0]hept-2-en-6-one by growing cultures of endophytic fungi. Adapted with permission from [128]. 2009. Elsevier.

Whole Cell Biocatalyst	Recovered Products			
	(*1S*,*5R*,*6S*)-*endo*-ol		(*1R*,*5S*,*6S*)-*exo*-ol	
	Yield %	e.e.%	Yield %	e.e.%
*Phomopsis* FE86	30	95	20	>99
*Pestalolta*	40	75	7	>99
*Phomopsis* F290	14	72	46	>99
*Epicoccum*	55	74	14	>99

**Table 12 molecules-28-07249-t012:** Time course of the biotransformation of (*rac*)-5-bromo-7-fluorobicyclo[2.2.1]-heptan-2-one by a fractured cell preparation of *Acinetobacter* NCIB 9871 resulting in the transitory production of the corresponding (*rac*)-*endo*-alcohol followed by the progressive accumulation of the corresponding (-)-(*1S*,*5R*)-2-oxa-lactone [2,77].

Sample Time (min) Post-Inoculation	% of the Assayed Sample (GC Peak Area)		
	Ketone	Alcohol	Lactone
30	85	12	0
60	65	31	2
90	44	51	3
120	39	56	3
150	20	75	4
300	60	25	13
600	55	20	23
960	50	6	43
1320	47	1	48

**Table 13 molecules-28-07249-t013:** Time course of the biotransformation of (*rac*)-5-bromo-7-fluorobicyclo[2.2.1]-heptanol by a fractured cell preparation of *Acinetobacter* NCIB 9871 resulting in the transitory production of the corresponding (*rac*)-ketone and the progressive accumulation of the corresponding (-)-(*1S*,*5R*)-2-oxa-lactone [2,77].

Sample Time (min) Post-Inoculation	% of the Assayed Sample (GC Peak Area)		
	Ketone	Alcohol	Lactone
30	7	90	1
60	36	59	3
90	64	29	4
120	72	21	6
150	76	12	9
300	79	5	13
600	55	3	23
960	52	2	40
1320	46	1	49

**Table 14 molecules-28-07249-t014:** Changes in the specific activity of both cyclohexanol dehydrogenase and cyclohexanone monooxygenase throughout the initial 8 hours of growth after inoculation of a culture of *Acinetobacter* NCIB 9871 into fresh cyclohexanol-based medium [198].

Time after Inoculation (h)	OD Culture (A500nm)	Specific Activity (U mg^−1^)	
		Cyclohexanol	Cyclohexanone
		Dehydrogenase	Monooxygenase
0.5	0.03	0.45	0.06
1	0.10	0.90	0.15
2	0.21	1.05	0.33
3	0.33	1.14	0.42
4	0.45	0.91	0.45
5	0.88	0.57	0.67
6	1.04	0.18	0.93
7	1.13	0	0.36
8	1.18	0	0.14

**Table 15 molecules-28-07249-t015:** Biooxygenation of (*rac*)-bicyclo[3.2.0]hept-2-en-6-one by purified preparations of 2,5-DKCMO and 3,6-DKCMO from *P. putida* ATCC 17453 [202]. Reprinted with permission from [212]. 1994. Royal Society of Chemistry.

Substrate	Enzyme	Products e.e.%		Products Ratio (Conversion %)
		(+)-3-oxa	(+)-2-oxa	
	2,5-DKCMO	>99	89	1:1.3 (100)
(*rac*)-bicyclo- [3.2.0]ketone				
	3,6-DKCMO	72	10	1:1.3 (100)

**Table 16 molecules-28-07249-t016:** Time course of the biotransformation of (*rac*)-6-*endo*-acetoxybicyclo[3.2.0]-hept-2-ene by a three-enzyme linear cascade (*P. cepacia* lipase + HLADH + MO1) [212].

% Composition of the Assayed Sample (GC Peak Area)				
Time (min)	6-*endo*-ester	6-*endo*-alcohol	Ketone	Lactones
0	100	0	0	0
30	55	24	16	4
60	52	16	11	19
90	52	11	8	29
120	51	2	4	41

**Table 17 molecules-28-07249-t017:** Time course for the repeat cycles of biotransformation of (*rac*)-bicyclo[3.2.0]hept-2-en-6-one in a membrane bioreactor by CHMO + FDH + PEG-NAD^+^ operating as a coupled-enzyme system. Reprinted with permission from [218]. 1996. Springer Nature.

Cycle	Time (Days)	Conversion (Total %)	e.e.% Lactones	
			3-oxa	2-oxa
1	1	>99	98	72
2	1	>99	97	72
3	1	97	97	71
4	1	89	95	72
5	1	75	97	70
6	1	60	95	68
7	1	38	94	69
**Recharge**				
8	1	92	95	70
9	1	68	94	68
10	1	50	93	63
11	1	38	93	62
12	1	21	92	60

**Table 18 molecules-28-07249-t018:** Time course of the repeat cycles of biotransformation of cyclohexanol in a membrane bioreactor charged with co-immobilised PDH and CHMO [272,273].

Cycle	Time (min)	Relative Activity (%) of the Co-Immobilised Enzymes (PDH + CHMO)
Pre-cycle	0	100
1	60	100
2	120	39
3	180	20
4	240	12
5	300	8

**Table 19 molecules-28-07249-t019:** Biooxygenation of (*rac*)-7,7-dichlorobicyclo[3.2.0]hept-2-en-6-one by various microbial BVMOs. Adapted from [1].

BVMO Tested	(*rac*)-Ketone Biotransformed (%)	Chiral 2-Oxa-lactone	
		Yield	e.e.%
CHMO_Arthro_ CHMO_Brevi-1_	89 69	59	n.d. n.d.
CHMO(MO14)	57	0	n.d.
CHMO_Rhodo_	100	25	>99

## Data Availability

Not applicable.

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
