# Peer review of "Bicyclo[3.2.0]carbocyclic Molecules and Redox Biotransformations: The Evolution of Closed-Loop Artificial Linear Biocatalytic Cascades and Related Redox-Neutral Systems"

_molecules, 2023, doi:10.3390/molecules28217249_

Round 1

Reviewer 1 Report

The manuscript reviews the evolution of bicyclic [3.2.0] carbon ring molecular oxidation and related cofactor cycles. Involves the evolution of the linear cascade from ADHs as a one-step biocatalyst for oxygen-functionalized bicyclic [3.2.0] carbon ring molecules to closed-loop artificial biocatalysis. Bicyclic [3.2.0] carbocyclic molecules are important in biosynthesis such as prostaglandins, as well as the concept of "dual intelligent common substrates" fusing biocatalytic cascades makes sense. However, the manuscript spans too long and the description is too verbose, suggesting that the content be streamlined and the focus be highlighted. In addition, there are too many references and are relatively old, and only 139 of the 316 articles are after 2000, and it is recommended to be simplified.

Author Response

I would like to thank Reviewer 1 for taking the time and interest to consider the manuscript.

The review focusses principally on a suite of interrelated studies that were conducted over the timescale 1991-1996, so the inclusion of a significant number of pre-2000 references is both to be expected and justifiable to explain the outcomes.

Like other examples of pioneering research that cannot be explained by conventional wisdom, progress was typically made in small increments that didn't always fit together initially, and which consequently require extensive post-factual rationalisation and explanation. Attempting to streamline the current text risks jeopardising its current sense of unity and cohesion.

Reviewer 2 Report

The manuscript is very important to the field and provides both historical and scientific overview of the cofactor recycling and its role in biocatalytic process development, in particular on the cascade reactions with application in industry. Thus, I recommend this paper for publication without revision.

Author Response

I would like to thank Reviewer 2 for taking the time and interest in the submitted manuscript. I much appreciate the comments made.

Reviewer 3 Report

Author has done an impressive work by preparing this review focused on the developments carried out for the biocatalytic modification of bicyclo[3.2.0]carbocyclic compounds, molecules presenting an overwhelming interest in organic chemistry as part of several natural and biologically active compounds. This a very comprenhensive review, including a huge amount of work, that covers from the seminal research in this field to the newest developments. So this review manuscript has to be published at Molecules, cause it will serve as a valuable guide to approach the chemistry of this type of compounds. Some minor comments on the revision will be:

- I miss a more concrete and detaliled information on alcohol dehydrogenases and Baeyer-Villiger monooxygenases, that are the main biocatalysts described herein. A brief introduction on both types of enzymes will be great.

- Figure 6 does not apport any revelant information and this is perfectly described in the text, so I would recommend to delete it.

-Please employ Baker yeast all over the manuscript.

- The names and possitions of some of the researchers is repeated several times all over the review, so I would recommend to delete some of them. For instance "Head of Chemical Research" at line 459, this is previously commented.

- I am missing a section for cyclohexanone monooxygenase, in the same way as sections 3.2. and 3.3. for other Baeyer-Villiger monooxygenases. CHMO is the most employed BVMO and requires a section for itself.

- Please use values of >99 instead of 100 when describing some of the results obtained. For instance, at Tables 7 and 8.

Author Response

I would like to thank Reviewer 3 for taking the time and interest to consider the manuscript. The following raised points are relevant:

1 The manuscript has been expanded accordingly to include more background information and relevant citations for both ADHs and Baeyer-Villiger monooxygenases (including specifically CHMO).

2. A relevant comment about the significance of Figure 6 has now been included in the reported outcomes of the Glaxo-based studies with HLADH (p10).

3. bakers has been amended to Bakers throughout the manuscript.

4. The references to research/academic positions of researchers has been minimised throughout the manuscript.

5. The included values of 100 in some of the Tables were as directly included in the relevant original publications, but I acknowledge that the current convention has changed, and so >99 has be substituted instead in each case.